# Neuronal differentiation and cell-cycle programs mediate response to BET-bromodomain inhibition in MYC-driven medulloblastoma

Pratiti Bandopadhayay et al.[#]

BET-bromodomain inhibition (BETi) has shown pre-clinical promise for MYC-amplified medulloblastoma. However, the mechanisms for its action, and ultimately for resistance, have not been fully defined. Here, using a combination of expression profiling, genome-scale CRISPR/Cas9-mediated loss of function and ORF/cDNA driven rescue screens, and cell-based models of spontaneous resistance, we identify bHLH/homeobox transcription factors and cell-cycle regulators as key genes mediating BETi's response and resistance. Cells that acquire drug tolerance exhibit a more neuronally differentiated cell-state and expression of lineage-specific bHLH/homeobox transcription factors. However, they do not terminally differentiate, maintain expression of CCND2, and continue to cycle through S-phase. Moreover, CDK4/CDK6 inhibition delays acquisition of resistance. Therefore, our data provide insights about the mechanisms underlying BETi effects and the appearance of resistance and support the therapeutic use of combined cell-cycle inhibitors with BETi in MYC-amplified medulloblastoma.

MYC-driven group 3 medulloblastoma is an aggressive pediatric brain tumor that is refractory to intensive multimodal therapy[1–3]. We and others have shown BET-bromodomain inhibition (BETi) to be a potential therapeutic strategy to target MYC-driven medulloblastomas and other cancers[4–9]. BET-bromodomain proteins bind to H3K27ac enhancers across the genome to recruit transcriptional complexes, thereby facilitating the expression of thousands of genes[10]. These include genes that regulate progression through the cell cycle and genes that mediate commitment of cell fate and differentiation in a context specific manner[4,10–14].

The transcriptional and phenotypic effects of BETi have been ascribed to reduced expression of MYC, cell-cycle regulators and stem-like transcriptional programs, in addition to global suppression of super-enhancer regulated genes[10]. However, it is currently unclear which of these BETi-modulated genes directly contribute to the growth-suppressing effects of BETi.

We hypothesized that genes that were required for medulloblastoma cell line growth and were sufficient to rescue the effects of BETi would represent key downstream effectors of this class of therapeutics. Here, we report a systematic approach to identify such genes using a combination of CRISPR/Cas9-based dependency screening, ORF/cDNA-mediated drug-rescue screens, and spontaneous models of drug resistance.

We found that BETi response is mediated by suppression of a combination of genes that regulate neuronal differentiation programs and of genes that regulate progression through the cell cycle. Furthermore, re-expression of these genes attenuates response to BETi by inducing a neuronally differentiated phenotype, while also maintaining proliferative capacity.

## Results

**A genomics approach to identify mediators of BETi response.** Noting that BET-bromodomain inhibitors mediate their effects through suppression of genes across the genome (Fig. 1, top panel), we sought to identify specific mediators of response to BET-bromodomain inhibition (BETi). We applied an integrative genomics approach using data from three sources: first, expression profiling of genes that are suppressed following BETi; second, genome-scale CRISPR/Cas9 screens to determine which of the suppressed genes are also essential for cellular viability; and third, a near-genome scale open reading frame (ORF) rescue screen to determine which of the suppressed genes are also sufficient to drive resistance to BETi (Fig. 1, bottom panel). In this way, we systematically evaluated which BETi target genes are both required for cellular proliferation, and able to rescue BETi phenotypes. We considered genes that were nominated by all three assays to be responsible for BETi-induced reductions in cell viability. We also validated the role of these genes and pathways in cells that acquired spontaneous tolerance to BETi.

**BETi generates widespread changes in expression.** We characterized the extent and uniformity of transcriptional effects of BETi in medulloblastoma models through expression profiling of four MYC-driven medulloblastoma cell lines treated with the BET-bromodomain inhibitor JQ1, relative to vehicle controls (Supplementary Data File 1 and Supplementary Figs. 1 and 2). The transcriptomic effects of BETi were widespread. For example, within the D458 and D283 cell lines, we respectively observed 5241 and 4762 genes to be downregulated following treatment with JQ1 (FDR ≤ 0.1) (Supplementary Data File 1). However, many of these expression changes are unlikely to affect cell survival, and therefore do not mediate BETi proliferation and viability effects.

**Genes suppressed by BETi tend to be cell-essential.** We therefore determined which of the suppressed genes are cell-essential. We applied a pooled CRISPR/Cas9 screen targeting 18,454 genes (Supplementary Fig. 3A) to each of the D458 and D283 cell lines. Cas9-expressing cells were infected with a genome-scaled pooled guide RNA (sgRNA) lentiviral library and passaged for 21 days. We considered genes that were depleted at the end of the assay relative to the early time point as cell-essential.

We identified 2455 and 2321 essential genes (Dependency score > 0.35 and FDR < 0.2 with no filtering of pan-essential genes; Supplementary Data File 2, Supplementary Figs. 4A, B, 5A, B), respectively. Of these, 876 (D458) and 760 (D283) genes were suppressed following BETi (Fig. 2a). The overlap between the two cell lines was significant (449 genes; $p < 0.0001$).

Across both lines, the essential genes exploited by BETi were enriched for members of 34 pathways, and most frequently included members of the cell-cycle (8 pathways), DNA replication/synthesis/elongation (5 pathways), and RNA processing/transcription pathways (5 pathways) (Supplementary Data File 3). One gene-set associated with MYC-activation was also enriched.

We validated these using two additional MYC-driven cell lines, D425, and D341. We applied the same genome-scale CRISPR screens to these lines, respectively and identified 2560 and 1980 essential genes (Supplementary Data File 2, Supplementary Figs. 4C, D and 5C, D). Among these, 1005 and 504 genes were among the 5000 genes that were most suppressed by JQ1 treatment (Supplementary Data File 1). This represented significant enrichment ($p = 0.003$ and $p < 0.0001$ respectively, Supplementary Fig. 3B). The 15 pathways that were most significantly suppressed by BETi ($p < 0.05$) included cell-cycle regulation (2 pathways), DNA replication (1 pathway) and MYC-activation (1 pathway), while another 3 pathways included members associated with chromatin regulation (Supplementary Data File 3).

We conclude that BETi consistently suppresses essential genes in sensitive MYC-amplified medulloblastoma cells, and that these genes consistently belong to cell cycle, DNA replication, and RNA processing pathways. However, the finding that these pathways are suppressed by BETi and are sufficient to generate cell death does not indicate that they are responsible for BETi's phenotypic effects. Some of these may represent essential pathways that are indirectly affected by BETi, after cell fate has already been determined.

**Essential genes that are suppressed by BETi.** We therefore attempted to narrow the set of essential genes and pathways that BETi exploits to those that are required for BETi phenotypic effects by determining, which genes were sufficient to rescue cells from BETi. We applied a lentivirally delivered ORF library encompassing 12,579 genes to both the D458 and D283 cell lines, each treated with either of two structurally distinct BET-bromodomain inhibitors (JQ1 and IBET151) or vehicle control (See Methods and Supplementary Fig. 3C). We measured the abundance of each ORF at initiation and completion of each assay to determine log-fold changes following BETi treatment.

For both cell lines, log-fold-changes for each gene were highly correlated between the IBET151 and JQ1 experiments (D458 $R^2$ = 0.76, $p$ value < 0.0001; D283 $R^2$ = 0.39, $p$ value < 0.0001) (Supplementary Fig. 3D, E), supporting similar target specificity for the two compounds.

In each cell line, we identified ORF constructs encoding 18 different genes that significantly rescued cells from either JQ1 or IBET151 and these lists were partially overlapping (31 genes total in both cell lines; Fig. 2b). We defined "rescue ORFs" as those conferring >1.5 log-fold enrichment with $q < 0.25$. The results for

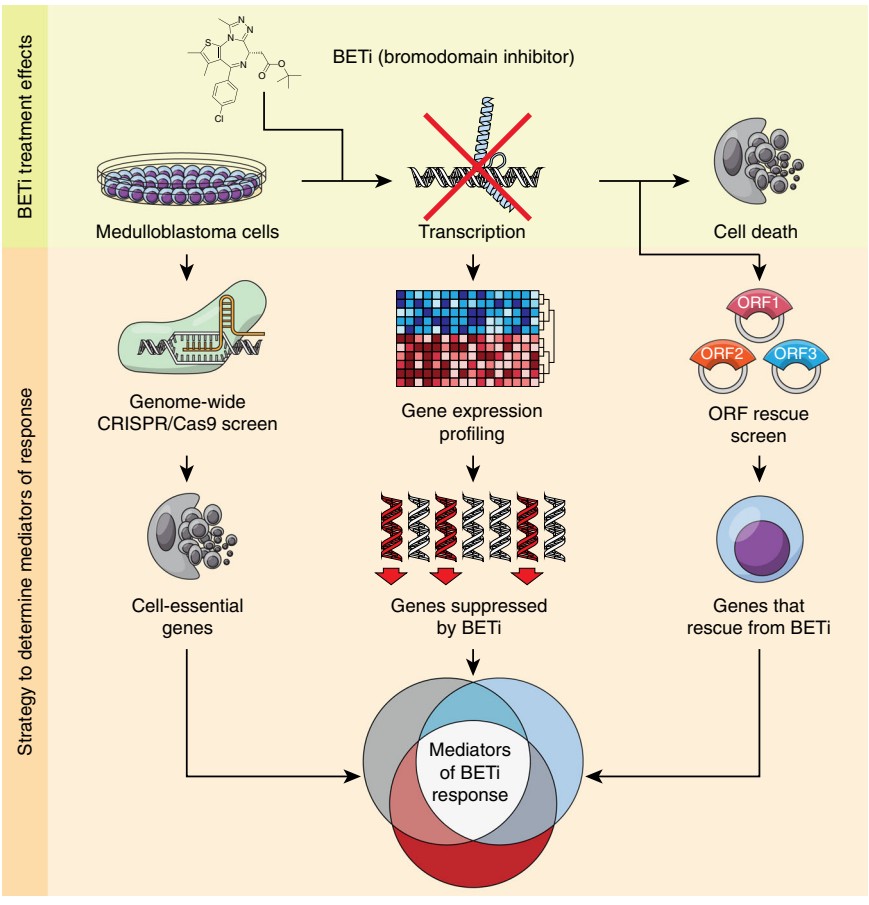

**Fig. 1** Integrative genomic analysis to identify mediators of BETi response. Schematic depicting overall analysis. Top panel: Treatment of medulloblastoma cells with BETi results in transcriptional suppression, contributing its phenotypic effects. Bottom panel: Integrative genomics approach incorporating genome-scale CRISPR-Cas9 screens to identify cell-essential genes, expression profiling following treatment with BETi to identify genes suppressed by BETi, and near genome-scale ORF rescue screens to identify genes that are required to be suppressed for BETi to exert its functional effect. The combination of all three assays identifies cell-essential genes that are suppressed by BETi, and whose suppression is necessary for decreased viability, thus mediating BETi response

the two cell lines overlapped extensively, as evidenced in three analyses. First, five of these rescue ORFs (*ATOH1*, *BCL2L1*, *BCL2L2*, *CCND3*, and *NEUROG1*) were common to both cell lines, a statistically significant overlap (*p* < 0.0001). Second, another four ORFs that met our significance threshold in only one cell line (*SPN*, *CCND2*, *NEUROG3*, and *MSX2*) all scored in the second line with a *q* < 0.25, but had fold-changes ranging from 1.15 to 1.4. Third, the five genes that scored among both cell lines included cell-cycle regulators and bHLH transcription factors that regulate neuronal differentiation: gene families that also include six of the 26 rescue ORFs that scored in only one of the lines (*CCND1*, *CCND2*, *NEUROG3*, *NEUROD6*, *NEUROD1*, and *NEUROD4*). Additional genes in these same families were identified using more relaxed cutoffs to define rescue ORFs (Supplementary Data File 4).

Proteins related to cell fate commitment, transcription, and developmental processes were significantly enriched in the rescue gene network (*q* < 0.0001), as were MYC-type basic helix-loop-helix (bHLH) (*q* < 0.001), cyclin (*q* < 0.01) and myogenic basic muscle-specific protein domains (*q* < 0.001, Supplementary Fig. 3F). Protein network analysis (performed using String, see Methods section) revealed that the pathway enrichment was also reflected by a high connectivity for the ORF network as a whole, with 42 edges (referring to protein–protein interactions; expected number of edges is 8) between the 31 nodes (individual ORFs) and a clustering coefficient of 0.749 (*p* < 0.0001, Fig. 2c).

Integrating all three datasets—gene expression, CRISPR/Cas9 screen, and ORF rescue—cell-cycle genes (*CCND2* and *CCND3*) scored in D458 and the anti-apoptosis gene *BCL2L1* and bHLH transcription factor-encoding gene *NEUROG1* scored in D283 (Fig. 2d). The cell-cycle gene *CCND2* also scored as an essential gene that is suppressed by JQ1 in D283 but only met the *q*-value (not log fold-change) threshold for a rescue gene.

We validated these genes in low-throughput assays (Fig. 3a, b, Supplementary Figs. 6A–C, 7). We overexpressed eGFP, *CCND2*, *CCND3*, *BCL2L1*, *MYOD1*, *MYOG*, *NEUROD1*, *NEUROG1*, and *NEUROG3*, in medulloblastoma cells and assessed proliferation in 1 μM of JQ1 or DMSO control. Overexpression of *CCND2*, *CCND3*, and *BCL2L1* rescued D458 cells from the effects of JQ1 (*p* values 0.002, 0.002, and 0.01) and *CCND3* and *NEUROG1* rescued D283 cells (*p* value = 0.002 and 0.01). There was a trend for overexpression of *CCND2* and *BCL2L1* in D283 to confer selective advantage in JQ1, but these did not reach statistical significance (*p* = 0.08 and 0.06, respectively). We also validated additional bHLH transcription factors as rescue genes: *MYOD1* in D458 (*p* = 0.04) and *NEUROD1* (*p* = 0.027), *NEUROG1* (0.02) and *NEUROG3* (*p* = 0.02) in D283. Overexpression of these ORFs did not confer growth advantages in any of the cell lines when passaged in DMSO (Supplementary Fig. 7). Expression of *BCL2L1* and *NEUROG3* attenuated JQ1-induced apoptosis relative to eGFP controls in both D458 and D283 (*p* values D458 *BCL2L1* 0.085 and *NEUROG3* 0.012; D283 *BCL2L1* <

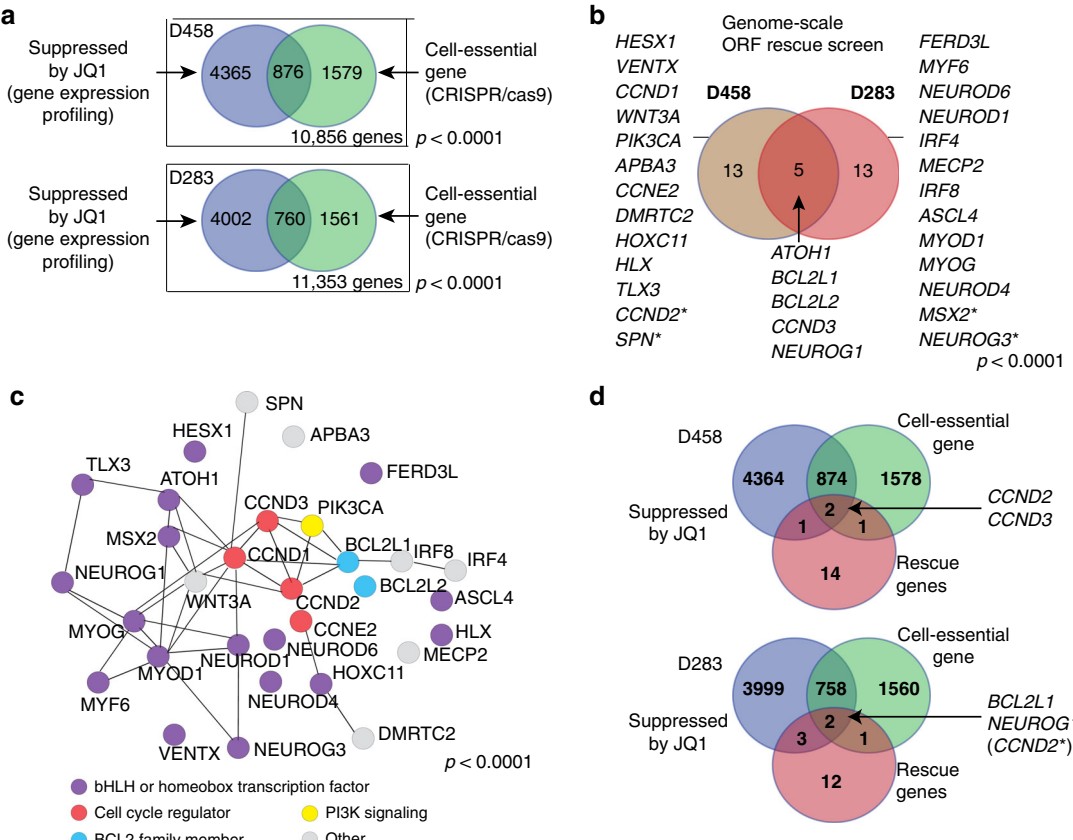

**Fig. 2** Rescue genes are enriched for cell-cycle regulators and bHLH/homeobox transcription factors. **a** Intersection of genes suppressed by 1 μM JQ1 with those identified to be cell essential (green) in D458 (top) and D283 (bottom). *p*-values indicate significance of overlap as determined with a Chi Square test. Source data: Supplementary Data Files 1 and 2. **b** Rescue genes identified in D458 (left) and D283 (right) cell lines following treatment with either JQ1 or IBET151. Asterisks indicate genes that scored as statistically significant rescue genes in both cell lines, but only met fold-change thresholds in the cell line shown. *p*-value indicates significance of overlap as determined by Fisher's Exact Test. Source Data: Supplementary Data File 4. **c** STRING[59] protein network analysis to identify direct and functional protein networks that exist between the entire set of candidate rescue ORFs identified across both cell lines. Protein-protein interactions (edges) between ORF rescue genes (nodes) that scored in either D458 of D283 following treatment with JQ1 or IBET151 are shown. *p* value indicates significance of enrichment of protein-protein interactions. Source Data: Supplementary Data File 4. **d** Venn diagram depicting overlap of genes that are suppressed by JQ1 (blue), score as dependencies in CRISPR-Cas9 screens (green) and are identified to be rescue genes (red) in D458 (top) or D283 (bottom). *CCND2 met both the *q* value threshold and the log-fold change threshold in D458, but only the *q*-value threshold in D283. Source data: Supplementary Data Files 1, 2, and 4

0.0001 and *NEUROG3* 0.0017, Fig. 2f), as did *CCND2* and *NEUROD1* in D283 (*p* values 0.0028 and <0.0001, respectively).

We also validated these ORFs as rescue genes in other patient-derived MYC-driven medulloblastoma cell lines: D341 and two that are passaged in serum-free conditions: CHLA01 and the recently generated cell line MB002. In each line, we found responses to JQ1 were attenuated by overexpression of a cell cycle regulator, BCL2 family member, and at least one bHLH/homeobox transcription factor (Supplementary Figs. 6C, 7). Thus, while we observed cell-specific differences in the magnitude of resistance for individual ORFs, we observed consistency at the pathway level (i.e., cell cycle, apoptosis avoidance, and bHLH/homeobox transcription factors).

BETi has been reported as a means to target MYC[4,6]. *MYC* was not included in the ORF screens. However, we previously demonstrated that ectopic MYC expression rescues D283 cells from BETi[6], and our analysis here confirmed *MYC* to be an essential gene (Supplementary Data File 2) that is transcriptionally suppressed by BETi in both D458 and D283 (Supplementary Data File 1)—indicating that MYC also fulfills all three criteria of a key essential gene that is suppressed by BETi. However, our

analysis indicates that *MYC* is not the sole mediator of BETi's phenotypic effects.

**Drug-tolerant D458 cells exhibit reversal of BETi effects.** We next sought to determine if the rescue genes identified in our ORF screens were differentially expressed in medulloblastoma cells that acquire BETi tolerance. We therefore passaged D458 cells and the related D425 line[15] in JQ1 until they exhibited growth in the presence of JQ1 and IBET151 (Supplementary Fig. 8A).

Drug-tolerant D425 and D458 cells maintained viability following treatment with JQ1, with reduced BETi-induced apoptosis and necrosis compared to drug naïve (or sensitive) control cells (Fig. 4a and Supplementary Fig. 8B, C), even when re-challenged with BETi after 30 days of drug withdrawal (Fig. 4b). We were unable to isolate drug-tolerant cells from the other medulloblastoma cell lines.

We hypothesized that drug-tolerant cells evade BETi effects by reversing its transcriptional consequences. In genome-scale expression profiles of sensitive and drug-tolerant cells following treatment with DMSO or JQ1, we found 3279 genes to be significantly upregulated in drug-tolerant cells cultured in JQ1

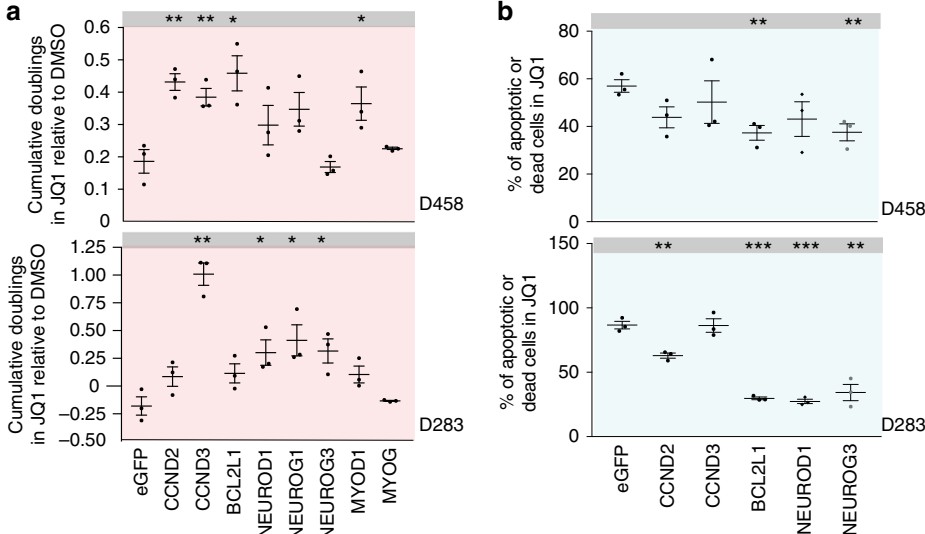

**Fig. 3** Expression of cell-cycle regulators, anti-apoptosis genes and bHLH/homeobox transcription factors rescue BETi effects **a** Low throughput rescue assays in D458 and D283 cells expressing eGFP, CCND2, CCND3, BCL2L1, NEUROD1, NEUROG3, MYOD1, or MYOG that were treated with JQ1 1 μM or DMSO control. Asterisks denote significant differences from eGFP controls (*$p < 0.05$, **$p < 0.01$, ***$p < 0.001$) as determined by Two-tailed unpaired $t$-tests. Error bars depict mean ± SEM. Source data: Source Data File. **b** Percentage (%) of apoptotic or dead cells in D458 and D283 cells expressing eGFP, CCND2, CCND3, BCL2L1, NEUROD1, or NEUROG3 treated with JQ1 2 μM for 72 h. Asterisks denote significant differences from eGFP controls (**$p < 0.01$, ***$p < 0.001$) as determined by two-tailed unpaired $t$-tests. Error bars depict mean ± SEM. Source data: Source Data File

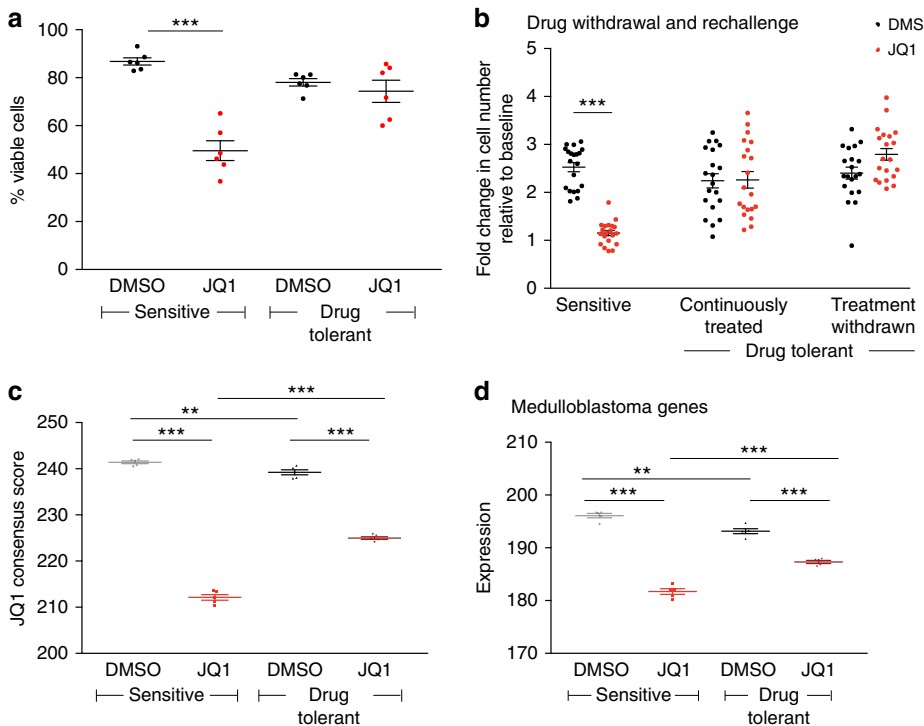

**Fig. 4** Drug-tolerant cells exhibit attenuated responses to BETi. **a** Percentage of viable cells among sensitive and drug-tolerant D425 and D458 populations after 72 h of treatment with JQ1. Error bars depict mean across six independent experiments ± SEM. ***$p < 0.001$ as determined by two-tailed unpaired $t$-tests. Source data: Source Data File. **b** Fold change in cell proliferation relative to pre-treatment baseline following 48 h of treatment with JQ1 among sensitive and drug-tolerant D425 and D458 cells that had been maintained in continuous JQ1 treatment ("drug-tolerant continuously treated") or in which JQ1 had been withdrawn for 30 days ("drug-tolerant treatment withdrawn"). Data depict 20 replicate measurements across 4 independent experiments; error bars depict mean ± SEM. Asterisks (***) depicts $p < 0.0001$ as determined by two-tailed unpaired $t$-test. Source data: Source Data File. **c** Expression of genes in the JQ1 consensus signature in sensitive and drug-tolerant D458 cells following 24 h of treatment with JQ1 (1 μM) or vehicle control. Data from five independent replicates are shown, error bars depict mean ± SEM. Asterisks denote significant differences (*$p < 0.05$, **$p < 0.01$, ***$p < 0.001$) as determined by two-tailed unpaired $t$-tests. Source data: Source Data File. **d** Expression of JQ1 medulloblastoma target genes in sensitive and drug-tolerant cells treated with JQ1 or vehicle control. Values represent mean of five replicate experiments ± SEM. Asterisks denote significant differences (*$p < 0.05$, **$p < 0.01$, ***$p < 0.001$) as determined by two-tailed unpaired $t$-tests. Source data: Source Data File

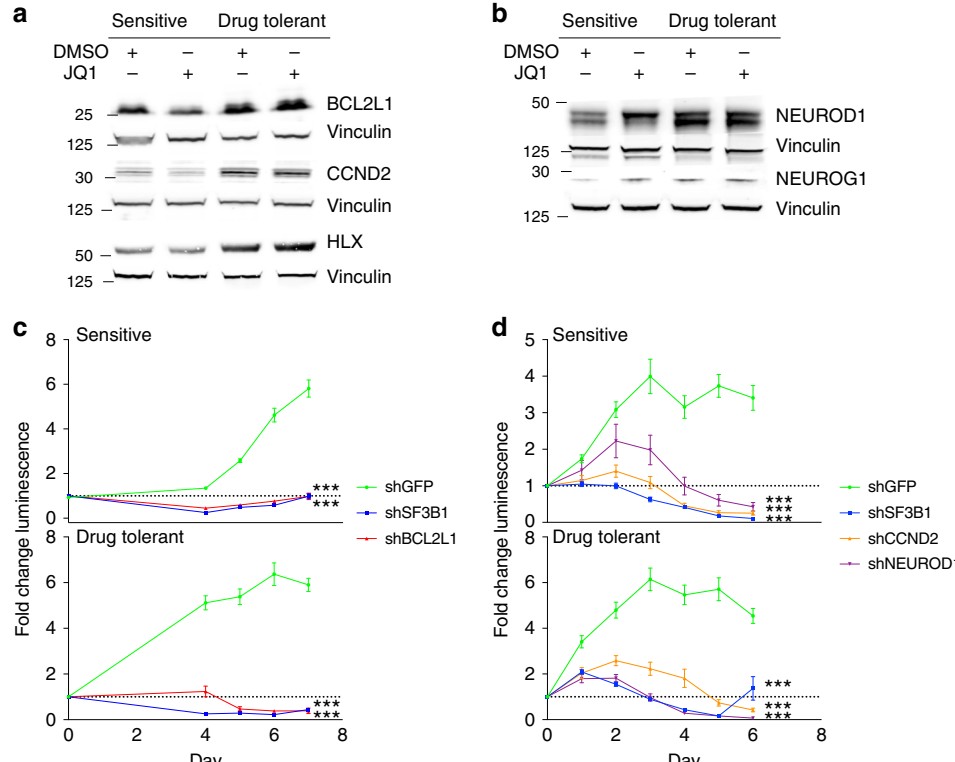

**Fig. 5** Drug tolerant cells express BETi mediator and rescue genes **a** Immunoblots probing for BETi-mediator proteins BCL2L1, CCND2, HLX in D458 sensitive and drug-tolerant cells following 24 h of treatment with JQ1 (1 µM) or vehicle control. Vinculin is included as a loading control. Source data: Source Data File. **b** Immunoblots probing for BETi-rescue proteins NEUROD1 and NEUROG1 in D458 sensitive and drug-tolerant cells following 24 h of treatment with JQ1 (1 µM) or vehicle control. Vinculin is included as a loading control. Source data: Source Data File. **c** Fold change in proliferation (as measured by luminescence) of D458 sensitive cells (top panel) and D458 drug tolerant medulloblastoma cells (bottom panel) following lentiviral infection with short hairpins targeting eGFP (negative control), SF3B1 (positive control) and BCL2L1. Values represent mean of 15 independent measurements across three replicate experiments ± SEM. Asterisks (***) denotes p values of <0.0001 as determined by two-tailed unpaired T-tests. Source data: Source Data File. **d** Fold change in proliferation (as measured by luminescence) of D458 sensitive cells (top panel) and D458 drug tolerant medulloblastoma cells (bottom panel) following lentiviral infection with short hairpins targeting eGFP (negative control), SF3B1 (positive control), CCND2 and NEUROD1. Values represent mean of 15 independent measurements across three replicate experiments ± SEM. Asterisks (***) denotes p values of < 0.0001 as determined by two-tailed unpaired t-tests. Source data: Source Data File

compared to drug-naïve cells (Supplementary Fig. 8D, Supplementary Data File 5A). These were significantly enriched for JQ1 transcriptional targets in D458 cells (1667 genes, p value < 0.0001). They were also enriched for genes that have previously been found to be suppressed by BETi both across cancers (JQ1 consensus signature[5,6] Supplementary Data File 5B, Fig. 4c) and among medulloblastoma cell lines[6] (Fig. 4d) (p value < 0.0001 in both cases). However, removal of JQ1 increased expression of both of these genesets in drug-tolerant cells even further (Fig. 3c, d, p < 0.0001 for both genesets), suggesting residual JQ1 activity. These cells also exhibited increased expression of BET-bromodomain target pathways including MYC activation and E2F signaling[4–6] and of MYC itself (Supplementary Figs. 9A, B, 11A). Taken together, these data suggest that the drug-tolerant D458 cells exhibit attenuated phenotypic and transcriptional responses to treatment with BETi.

We next validated that rescue genes identified in our ORF screens, including cell-cycle regulators and bHLH transcription factors, were relevant in these models. Among the 18 rescue ORFs, five ORFs (*BCL2L1*, *CCND2*, *HLX*, *NEUROD1*, and *NEUROG1*) were either re-expressed in drug tolerant cells following suppression with BETi or exhibited increased expression with drug-tolerance. The cell-cycle regulator *CCND2* was suppressed by BETi (p value 0.04, Fig. 5a and Supplementary

Fig. 9C), while the bHLH transcription factor *HLX* and the anti-apoptotic protein *BCL2L1* trended towards suppression with BETi (p value 0.05 and 0.059, respectively, Fig. 5a and Supplementary Fig. 9C–E). All three of these were re-expressed at both the mRNA and protein levels in drug-tolerant cells (Differential mRNA expression q < 0.1, Fig. 5a and Supplementary Figs 9C–E, 11B (mRNA), p values for proteins 0.0002, 0.02, and 0.01, respectively), suggesting that these were "mediator" genes whose expression had been reinstated. The other 15 rescue ORFs had not been suppressed by BETi, but expression of two of these also increased in drug-tolerant cells: the bHLH transcription factors *NEUROD1* and *NEUROG1* (Fig. 5b, Supplementary Fig. 9F, G, p values 0.0005 and 0.002, respectively for sensitive and drug-tolerant cells in DMSO).

To further evaluate the functional significance of the anti-apoptosis, cell cycle and bHLH transcription factors in drug-tolerance, we suppressed *BCL2L1*, *CCND2*, and *NEUROD1* using short hairpin RNAs, confirming each of these genes to be essential in both D458 drug-naïve and drug-tolerant cells (p values < 0.0001 in all cases, Fig. 5c, d and Supplementary Fig. 11C, D). We concluded that rescue genes from cell-cycle, bHLH transcription factor, and anti-apoptotic pathways are cell-essential in D458 cells and are re-expressed in cells that acquire drug-tolerance, remaining genetic dependencies in those cells.

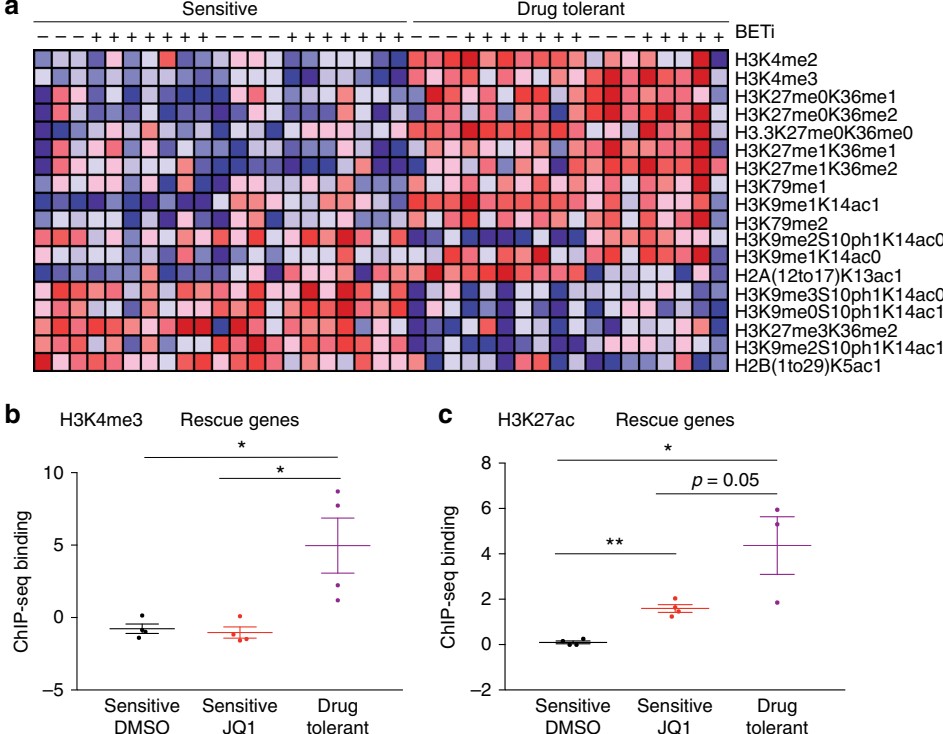

**Fig. 6** Drug-tolerant cells exhibit changes in chromatin landscape and activating marks preferentially bind to rescue genes. **a** Heatmap representing abundances of differentially altered chromatin marks as identified by global chromatin profiling of sensitive and drug-tolerant D425 and D458 cells passaged in DMSO or BETi (JQ1 and IBET151) for 24 h. Red and blue respectively represent values above and below the median across each row. Source data: Supplementary Data File 8. **b, c** ChIP-seq binding scores (*Z*-transformed) of **d** H3K4me3 and **e** H3K27ac marks to rescue genes identified in ORF screens in sensitive cells treated with DMSO (*n* = 2) or 1 μM JQ1 (*n* = 2) and drug-tolerant cells passaged in 1 μM JQ1 (*n* = 4). Error bars depict mean ± SEM. Asterisks denote significant differences (*$p < 0.05$, **$p < 0.01$, ***$p < 0.001$) as determined by two-tailed unpaired *t*-tests. Source data: Source Data File

**Drug-tolerant cells exhibit altered chromatin landscape**. We next explored possible mechanisms by which drug-tolerant cells express rescue genes including the cell-cycle regulator *CCND2* and bHLH/homeobox transcription factors. We first explored genetic mechanisms by performing whole-exome sequencing of drug-tolerant D458 and D425 cells relative to matched drug-sensitive controls. We found no mutations that recurred across two or more drug-tolerant replicates that were not present in drug-sensitive cells, nor any activating mutations or copy-number alterations in BET-bromodomain containing genes, rescue genes, or their associated pathways (Supplementary Data Files 6 and 7). These results suggest that the drug-tolerant cells express these genes through alternative mechanisms.

In contrast, drug-tolerant cells exhibited changes in their chromatin landscape. We profiled global histone marks using a targeted, quantitative mass spectrometry approach[16] in sensitive and drug-tolerant D425 and D458 cells, in both the presence and absence of BET-bromodomain inhibitors (Fig. 6a and Supplementary Data File 8). Of the 76 histone marks included in the assay, 18 were differentially altered in drug-tolerant lines, 12 of which were up-regulated (Supplementary Data File 8). Drug-tolerant cells were enriched for methyl marks and for those that facilitate transcription ($p < 0.05$ in both cases). These include the promoter-associated mark H3K4me3 (and the related mark, H3K4me2), H3K9me1 (which has been associated with active enhancers[17]), and upregulation of H3K79me1, H3K79me2, and H3.3, all of which have been described to be associated with transcription[18–20]. In contrast, drug sensitive cells had increased levels of the polycomb repressive marks H3K27me3 and the H3K9me3. We did not observe any statistically significant

differences in the H3K27ac profiles of drug-sensitive and drug-tolerant cells. Taken together, these data suggest that, in the setting of BETi, drug-tolerant cells maintain gene transcription through upregulation of activating methyl marks and down-regulation of repressive marks.

These changes in chromatin landscapes appear to facilitate the expression of rescue genes that we had observed in our drug-tolerant cell lines (and had scored in the D458 rescue screen, including cell-cycle regulators and bHLH/homeobox transcription factors). We performed ChIP-seq for the promoter-associated and enhancer-associated marks H3K4me3 and H3K27ac, respectively, in sensitive cells treated with DMSO or JQ1 and in drug-tolerant cells treated with JQ1. Rescue genes exhibited increased levels of total H3K4me3 in drug tolerant D458 medulloblastoma cells compared to sensitive cells in either DMSO (*p* value 0.03) or JQ1 (0.02, Fig. 6b). Treatment of D458 cells with JQ1 was associated with increased levels of total H3K27ac at rescue genes (*p* value 0.0002, Fig. 6c), persisting in drug tolerant cells, which maintained elevated levels of total H3K27ac at rescue genes relative to untreated D458 cells (*p* value 0.01, Fig. 6c). There was a trend for D458 drug-tolerant cells to exhibit increased levels of H3K27ac binding at rescue genes compared to sensitive cells treated with JQ1, but this did not reach statistical significance (*p* value 0.05, Fig. 6c). We did not observe similar changes in relation to genes that did not score as rescue genes in our ORF screens (Supplementary Fig. 11E). In aggregate, these data suggest that drug tolerance involves activation of promoter-associated and enhancer-associated marks preferentially for rescue genes.

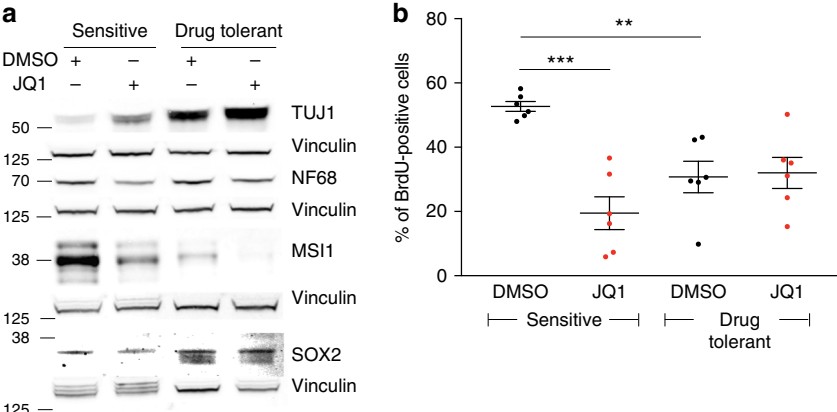

**Fig. 7** Drug tolerant cells exhibit altered cell state. **a** Immunoblots for neuronal differentiation and stem markers in sensitive and drug-tolerant D458 cells in the presence of JQ1 (1 μM) or DMSO control. Source data: Source Data File. **b** Percentage of BrdU positive D425 and D458 sensitive and drug-tolerant cells following 72 h of treatment with JQ1 (1 μM) or DMSO control. Data represent three independent experiments per cell line; error bars depict mean ± SEM. Asterisks denote significant differences (*$p < 0.05$, **$p < 0.01$, ***$p < 0.001$) as determined by two-tailed unpaired $t$-tests. Source data: Source Data File

**Changes in cell-state and differentiation attenuate response to BETi.** The findings that expression of bHLH transcription factors and altered chromatin landscapes were both associated with drug tolerance led us to hypothesize that the drug-tolerant cell lines exhibit altered differentiation states. bHLH transcription factors have been reported to play an essential role in neural development and regulation of cell-fate commitment[21,22]. We confirmed overexpression of *NEUROD1* and *NEUROG1* to be associated with an increased drive towards neuronal differentiation, as measured by levels of the neuronal marker TUJ1 (TUBB3) (Supplementary Figs. 9H, 12A, $p$ values 0.03 and 0.02, respectively).

Drug-tolerant D458 cells exhibited changes in differentiation status relative to sensitive cells (Figs. 7a, 5b and Supplementary Figs. 9, 10), with increased expression of neuronal markers NEUROD1 ($p$ value < 0.001 sensitive and drug tolerant cells in DMSO), NEUROG1 ($p$ value 0.002 sensitive and drug tolerant cells in DMSO), TUJ1 (0.049 sensitive and drug tolerant cells in DMSO and <0.0001 between sensitive cells treated with DMSO or JQ1) and NF68 ($p$ value < 0.0001 between sensitive cells treated with DMSO or JQ1). Drug-tolerant cells also exhibited suppression of the stem marker MSI1 ($p$ value 0.002 sensitive and drug tolerant cells in DMSO).

The finding that expression of bHLH genes leads to both differentiation and resistance raises a conundrum. While stem cells exhibit self-renewal capacity and progression through S-phase, cells on the path towards neuronal differentiation exit S-phase, with cell-cycle arrest corresponding with terminal differentiation[23]. However, we found that BETi drug-tolerant cells continue to cycle through S-phase, even in the presence of BETi, albeit at lower rates than untreated drug-naïve controls. Treatment of drug-sensitive D425 and D458 with JQ1 results in significant reductions in the percentage of BrdU positive cells relative to those treated with DMSO vehicle controls (19.5 % vs. 53%, $p$ value < 0.0001, Fig. 7c). Drug-tolerant D425 and D458 cells do not exhibit further changes in the proportion of BrdU positive cells when challenged with JQ1 or DMSO, with approximately 30% of cells staining positive for BrdU in both conditions. However, the percentage of BrdU positive drug tolerant cells in DMSO is significantly lower than sensitive cells (31% vs. 53%, $p$ value 0.0017, Fig. 7c).

This conundrum may explain why cell-cycle regulators were also prominent "hits" in our integrative analysis (Figs. 1c, 2c, d) and why *CCND2*, a transcriptional target of BETi, was re-expressed in drug-tolerant cells (Fig. 5a). Thus, we conclude that drug tolerant cells have acquired a cellular state that is primed for differentiation (e.g., via upregulation of bHLH) and exhibits global alterations in chromatin structure, but paradoxically maintains its ability to progress through the cell cycle (e.g., by upregulation of cell cycle genes).

**Inhibition of cell-cycling delays acquisition of resistance to BETi.** Our finding that drug-tolerant cells exhibited upregulation of the cell-cycle regulator (and BETi rescue gene) *CCND2* and continue to cycle through S-phase led us to hypothesize that cell-cycle inhibition would prevent the acquisition of drug tolerance[24].

We first evaluated the acute efficacy of the combination of JQ1 with the CDK4/CDK6 inhibitor LEE011[25]. We found this combination to meet BLISS synergy criteria[26] in D458, MB002, and D341 (Supplementary Fig. 12B–F). We further validated synergy (using the LOEWES model for synergy) between LEE011 and JQ1 across a wider range of concentrations in D458 ($p$ value < 0.0001), MB002 ($p$ value < 0.0001) and D283 ($p$ value < 0.0001), along with an additional MYC-driven line HD_MB003 ($p$ value < 0.0001, Supplementary Figs. 13, 14). With prolonged treatment of D458 cells, we also found the addition of LEE011 to JQ1 delayed the acquisition of drug tolerance relative to treatment with JQ1 alone (Fig. 8a).

Combined treatment with LEE011 and JQ1 also attenuated the acquisition of resistance in vivo. In mice harboring flank injections of D458 and MB002 (Fig. 8b, c). GSEA revealed seven pathways to be significantly downregulated in RNA from D458 tumors treated with combination therapy compared to those treated with JQ1 alone, including two associated with the cell cycle (E2F_Targets and G2M_Checkpoint), and two enriched with MYC targets (Fig. 5d), suggesting addition of LEE011 effectively suppressed the cell cycle. In mice bearing intracranial xenografts of D458 cells, treatment with LEE011 alone had no effect on tumor growth or survival, while single agent JQ1 reduced tumor growth ($p = 0.0004$ on day 14) and prolonged survival ($p = 0.0002$) relative to vehicle controls. However, the combination of LEE011 and JQ1 further attenuated tumor growth compared to those treated with JQ1 alone, with significantly less bioluminescence after one month (Supplementary Fig. 14C $p$ < 0.05) and increased overall survival (Fig. 9a, $p < 0.05$). We observed similar synergy in two additional patient-derived *MYC*-driven intracranial medulloblastoma models: Med-114FH (Fig. 9b) and Med-411FH (Fig. 9c), which exhibited significantly prolonged survival when treated with both LEE011 and JQ1 compared to vehicle controls ($p$ values 0.01 and 0.02) or

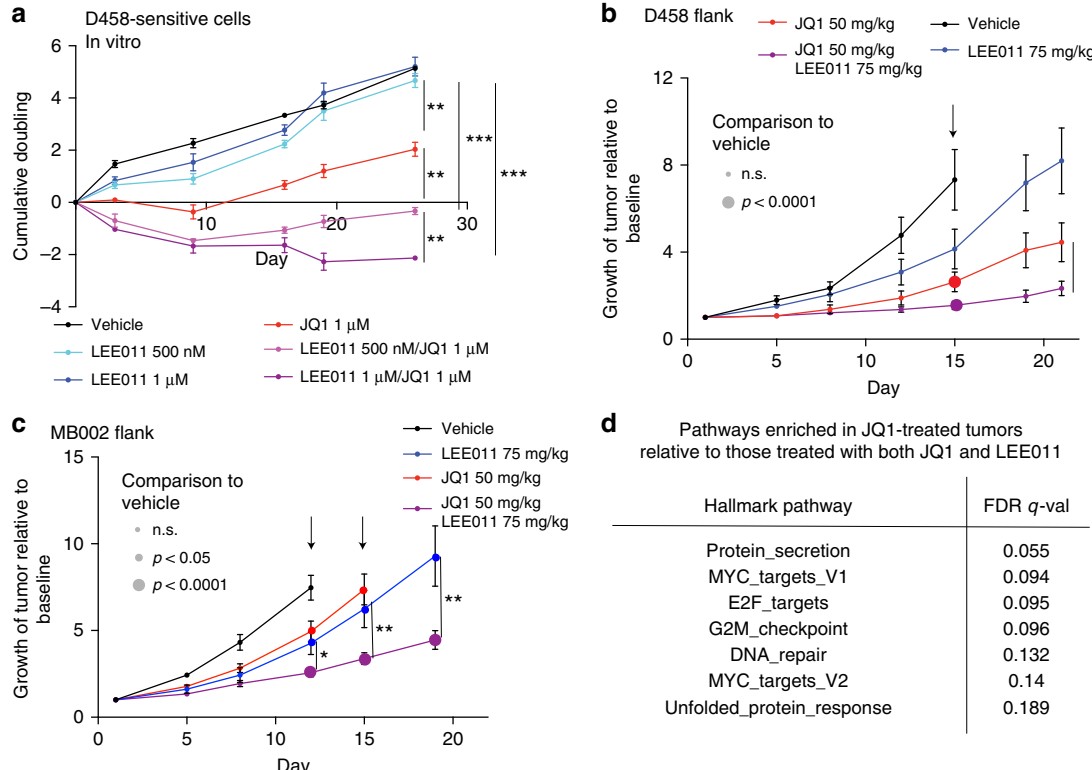

**Fig. 8** Combination of BETi and CDK4/CDK6 attenuates resistance in flank models **a** Cumulative doubling of D458 cells passaged in DMSO, LEE011 (500 nM or 1 μM), JQ1 (1 μM) or the combination of JQ1 (1 μM) with LEE011 at either 500 nM or 1 μM. Data from three independent replicates are shown, error bars depict mean ± SEM. For all panels, asterisks denote statistically significant differences (*$p < 0.05$, **$p < 0.01$, ***$p < 0.001$) as determined by two-tailed unpaired $t$-tests. Source data: Source Data File. **b** Growth of D458 flank xenografts treated with vehicle, JQ1 (50 mg/kg/day), LEE011 (75 mg/kg/day) or the combination of both compounds (JQ1 50 mg/kg/day and LEE011 75 mg/kg/day). Three vehicle treated mice were euthanized on day 15 due to their tumor size (shown by arrow). Data represent mean tumor volume ($n = 10$), normalized to day 1, ± SEM. Large circles denote time points with significant ($p < 0.0001$) improvement in tumor volume relative to DMSO controls. Asterisk denotes significant ($p < 0.05$) improvement in tumor volume among mice treated with JQ1 and LEE011 relative to those treated with JQ1 alone as determined by two-tailed unpaired $t$-tests. Source data: Source Data File. **c** Growth of MB002 flank xenografts treated with vehicle, JQ1 (50 mg/kg/day), LEE011 (75 mg/kg/day or the combination of both compounds (JQ1 50 mg/kg/day and LEE011 75 mg/kg/day). Arrows depict days in which mice were euthanized due to their tumor size. Data represent mean tumor volume ($n = 10$), normalized to day 1, ± SEM. Medium and large circles denote significant ($p < 0.05$ and $p < 0.0001$, respectively) differences between treated tumors at each time point relative to DMSO controls. Asterisks denote statistically significant improvements in tumor volume among mice treated with JQ1 and LEE011 relative to those treated with JQ1 alone (*$p < 0.05$, **$p < 0.01$, ***$p < 0.001$) as determined by two-tailed unpaired $t$-tests. Source data: Source Data File. **d** Gene sets enriched in mRNA extracted from in vivo D458 flank xenografts treated with JQ1 relative to xenografts treated JQ1 and LEE011 (and thus suppressed by the combination of JQ1 and LEE011)

treatment with JQ1 alone ($p$ values 0.017 and 0.007). No differences were observed with either single agent JQ1 or LEE011 compared to vehicle controls.

Taken together, these in vitro and in vivo experiments across a number of cell lines indicate synergy between JQ1 and LEE011 in BETi naïve cells, with the addition of LEE011 delaying the acquisition of resistance.

**Drug-tolerant cells occupy a mixed differentiation state.** Our findings suggest a model whereby BETi suppresses essential bHLH lineage-specific transcription factors only in relatively undifferentiated cells, while more neuronally differentiated medulloblastoma cells maintain expression and viability in the presence of drug (Fig. 10a). However, we find that the cells do not terminally differentiate, maintaining expression of SOX2 (Fig. 7a and Supplementary Fig. 10) and re-expressing cell-cycle regulators such as CCND2 to promote cell-cycle progression (Fig. 5a). We therefore hypothesized that resistance accrues in populations of medulloblastoma cells that are somewhat, but incompletely, neuronally differentiated (Fig. 10a).

Given that sensitivity to BETi is related to neuronal differentiation within cell lines, we reasoned that across cell lines, those with increased expression of neuronal differentiation markers would be inherently less sensitive to BETi. To test this, we interrogated the CTRP dataset[27,28] and found Tuj1 expression to be correlated with resistance to JQ1 across 783 cancer cell lines (Supplementary Fig. 14C). Within lineages, expression of Tuj1 was associated with attenuated responses to JQ1 in cell lines derived from the central nervous system ($p = 0.044$, Fig. 10b), prostate ($p = 0.047$, Supplementary Fig. 14D) and thyroid ($p = 0.012$, Supplementary Fig. 14E. Both prostate and thyroid cancers have been reported to express Tuj1 (in addition to several other markers of neuroendocrine differentiation)[29,30]. These results suggest that the relevance of neuronal differentiation markers to BETi sensitivity are not restricted to D283 and D458 cells but generalize across cancer types that express these markers.

We next sought to characterize the heterogeneity of stem cell and lineage markers within human medulloblastomas. We profiled Tuj1 expression, in addition to the stem markers Nestin, Olig2 and Sox2 across a panel of 46 medulloblastomas. There was heterogeneity in the proportion of Tuj1 expressing cells both

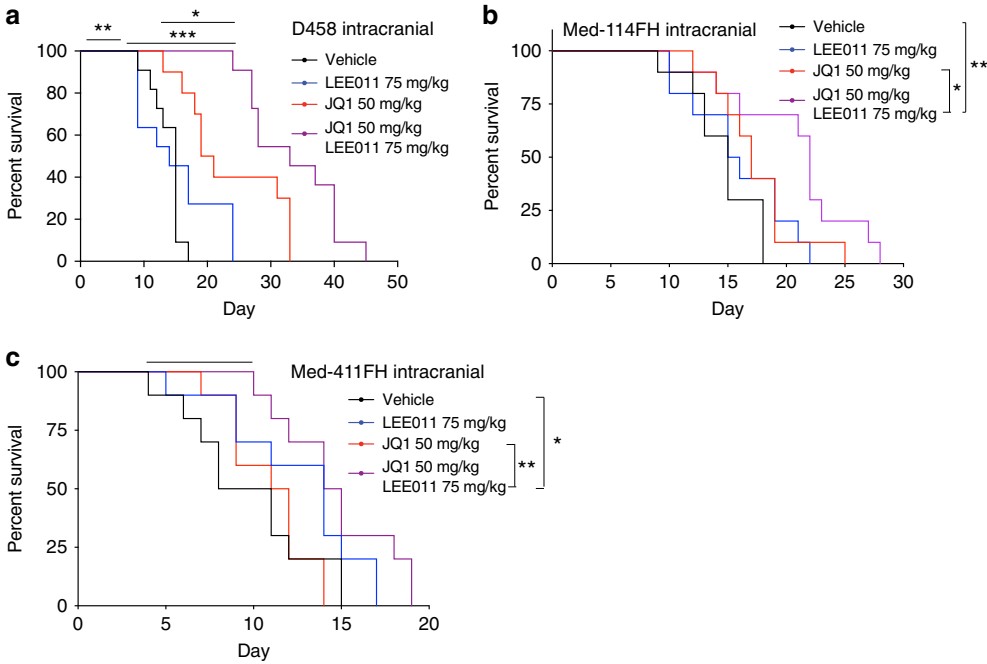

**Fig. 9** CDK4/6 inhibition attenuates resistance to BETi in intracranial models **a** Kaplan–Meier survival curves of mice bearing intracranial D458 xenografts treated with vehicle, JQ1 (50 mg/kg/day), LEE011 (75 mg/kg/day) or the combination of both compounds (JQ1 50 mg/kg/day and LEE011 75 mg/kg/day). Asterisks denote statistically significant differences (*$p < 0.05$, **$p < 0.01$, ***$p < 0.001$) as determined by log-rank (Mantel-Cox) tests. **b** Kaplan–Meier survival curves of mice bearing intracranial Med114 xenografts treated with vehicle, JQ1 (50 mg/kg/day), LEE011 (75 mg/kg/day) or the combination of both compounds (JQ1 50 mg/kg/day and LEE011 75 mg/kg/day). Asterisks denote statistically significant differences (*$p < 0.05$, **$p < 0.01$, ***$p < 0.001$) as determined by log-rank (Mantel-Cox) tests. **c** Kaplan–Meier survival curves of mice bearing intracranial Med411 xenografts treated with vehicle, JQ1 (50 mg/kg/day), LEE011 (75 mg/kg/day) or the combination of both compounds (JQ1 50 mg/kg/day and LEE011 75 mg/kg/day). Asterisks denote statistically significant differences (*$p < 0.05$, **$p < 0.01$, ***$p < 0.001$) as determined by log-rank (Mantel-Cox) tests

within and across tumor samples (Supplementary Data File 9). Across all tumors, medulloblastomas harbored a subpopulation of cells (median 9%, range 0–86%) that expressed Tuj1 but were negative for the stem cell markers, thus representing a more differentiated phenotype (Fig. 10c). In addition, we also observed subpopulations of cells that exhibited a mixed phenotype with co-expression of the neuronal marker TUJ1 with the stem markers.

**Tolerance to BETi is predetermined.** The heterogeneity within medulloblastomas led us to hypothesize that pre-existing sub-populations of differentiated neuronal cells may be selected for by BETi due to their inherent resistance. We explored this using two approaches. First, using flow cytometry, we observed an enrichment of Tuj1 positive and MSI1 negative drug-tolerant D458 cells with BETi. Flow cytometry analysis revealed 0.7% of cells to Tuj1 positive while negative for the stem marker MSI1 (thus repre-senting a well-differentiated subpopulation) in untreated D458 cells. This population significantly expanded to 10% of drug-tolerant cells in JQ1 (Fig. 10d).

Second, we leveraged barcoding technology to confirm that the changes in population structure observed in D458 cells reflected selection of subpopulations of cells that were predetermined to tolerate BETi. We used a lentivirally delivered library comprising 600,000 DNA barcodes to individually label both D283 and D458 cells and determine whether their progeny consistently survived treatment across eight replicate experiments (Fig. 10e). Only 23 and 25% of all barcodes detected across all conditions were present after treatment with JQ1, in D283 and D458 cells respectively. Among these, 48 and 8% were present post-treatment across all replicates (Fig. 10f and Supplementary Fig. 14F). These specific barcodes were shared with the other JQ1-treated replicates but not with the DMSO controls

(Supplementary Fig 13G, 13H). The enrichment of the same barcodes following treatment with JQ1 in replicate experiments supports the presence of a population of cells that are predetermined to survive treatment.

## Discussion
Small molecule inhibitors of transcriptional modulators are increasingly showing preclinical promise across a range of can-cers. However, these compounds have pleiotropic effects, making it difficult to identify the genes and pathways that mediate their efficacy. Identifying genes that are suppressed, and whose sup-pression is both necessary and sufficient to generate specific phenotypes, has been used as an approach to detect mediators of mechanistic effects for decades[31,32]. With the advent of high-throughput functional screening including genome-scale CRISPR and ORF rescue screens, it is now possible to test sufficiency and necessity on a genome-scale basis, which is the approach we have taken here. Similar integrative genomic approaches may also be effective in clarifying the primary targets of other modulators of transcription such as inhibitors of chromatin/transcriptional complexes[33,34].

Our analysis support heterogeneity within (and between) medulloblastomas, a finding that has also been reported in single-cell RNA-sequencing studies of medulloblastomas[35]. We identi-fied cells that express both genes associated with neuronal dif-ferentiation and stemness as being associated with drug-tolerance. Similar populations of cells have been identified in other normal developmental hierarchies[36–37], and in cancer, and have been termed transit amplifying cells[38,39] or in the neural context, "activated quiescent neural progenitor cells". These cells have been shown to harbor a phenotype that is more differentiated than stem cells, and to harbor increased proliferative potential

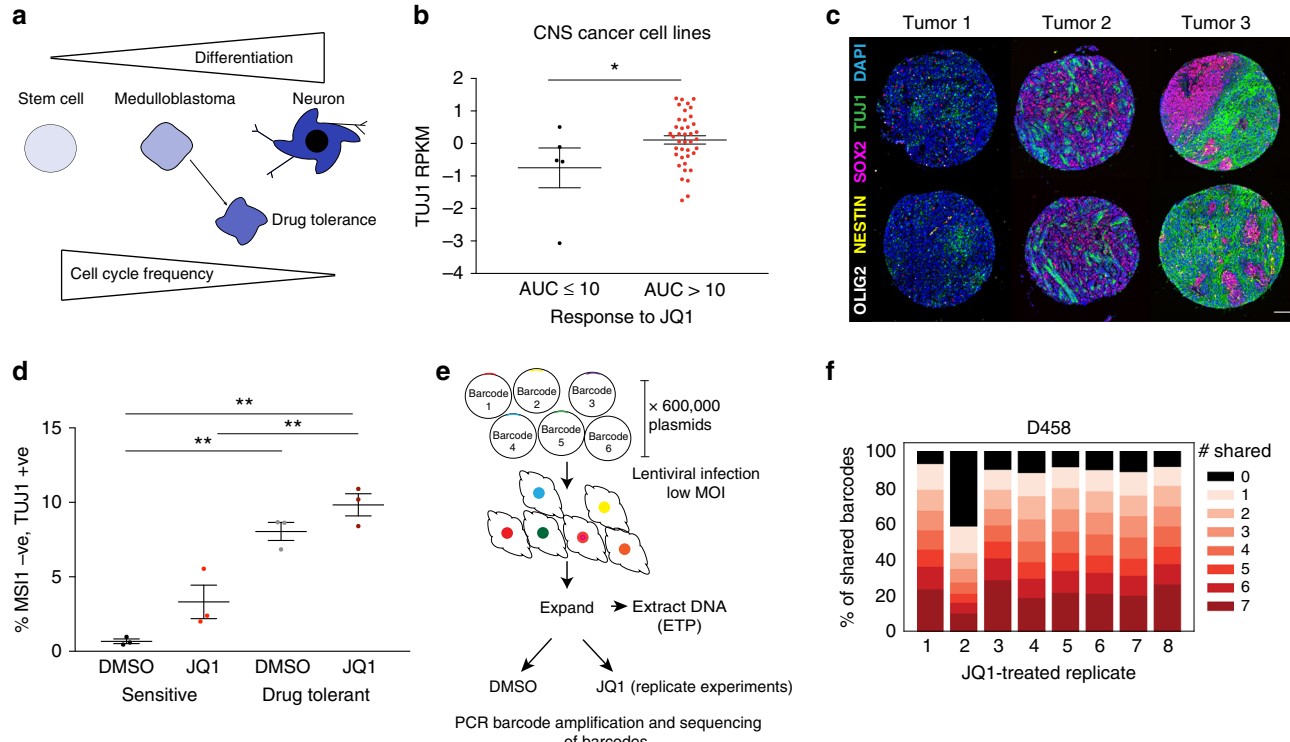

**Fig. 10** Drug-tolerant cells exhibit an altered differentiation state that is also present in human medulloblastoma. **a** Model for altered differentiation in BETi-tolerant medulloblastoma cells. Human medulloblastomas express both stem-cell and neuronal differentiation markers. Following treatment with BETi, drug-tolerant cells exhibit a more neuronal phenotype however do not undergo terminal differentiation, maintaining the ability to progress through the cell-cycle. **b** Expression levels of TUJ1 (TUBB3) and JQ1 sensitivity of 46 central nervous system cell lines from the CTRP database. AUC denotes area under the curve. Asterisks denote statistically significant differences (*$p < 0.05$, **$p < 0.01$, ***$p < 0.001$) as determined by a two-tailed unpaired $t$-test. Source data: Source Data File. **c** Representative images of three medulloblastomas stained for TUJ1 (green), SOX2 (red), NESTIN (yellow), OLIG2 (white) and DAPI (blue). Scale bar is 100 µm). Source data: Supplementary Data File 9. **d** Percentage of D458 sensitive and drug-tolerant cells that are TUJ1 positive and MSI1 negative in the presence of JQ1 (1 µM) or DMSO control. Data from three independent experiments are shown; error bars depict mean ± SEM. Asterisks denote statistically significant differences (*$p < 0.05$, **$p < 0.01$, ***$p < 0.001$) as determined by a two-tailed unpaired $t$-test. Source data: Supplementary Data File 9. **e** Schematic depicting barcoding experiments with 600,000 barcode library. A pooled lentiviral library encompassing 600,000 unique DNA barcodes was used to label individual cells. Cells were expanded before being placed in replicate DMSO or JQ1 drug-treatment experiments. Barcodes were PCR amplified prior to treatment (ETP = early time point) and from replicate experiments at the drug of treatment and subjected to next-generation sequencing. **f** Percentage of shared DNA barcodes across replicates of barcoded D458 cells treated with JQ1 (2 µM). Within each replicate, the number of other replicates with whom each barcode is shared is depicted. Source data: Source Data File

compared to quiescent stem cells or terminally differentiated neurons. Our findings suggest targeting the BETi-tolerant cell-states may represent a potential therapeutic strategy to optimize BETi efficacy.

Furthermore, the observation that BETi-tolerance is pre-determined in medulloblastoma cells provides a rationale for initiating combination therapies to ablate these subpopulations of cells in BETi treatment naïve cells. We observe BETi combination treatment with the CDK4/6 inhibitor LEE011 to attenuate the acquisition of drug-tolerance. Cell-cycle inhibition with CDK4/6 inhibitors including palbociclib have been shown to have pre-clinical promise in models of MYC-amplified medullo-blastoma[40–42]. The combination of CDK4/6 and BET-bromodomain inhibition has also been reported to be synergis-tic in other contexts[43], while in medulloblastoma, BETi has also been shown to synergize with inhibition of CDK2[44].

We have also identified bHLH/homeobox transcription factors and BCL2 family members as mediating response and resistance to BETi. Future work will examine whether also inhibiting these pathways may represent a therapeutic strategy to attenuate resistance to the combination of cell-cycle inhibition and BETi. While our integrative analysis has highlighted multiple mediators of BETi response, *MYC* remains an important target. We found

*MYC* to be re-expressed in drug-tolerant cells, and for rescue ORFs to be enriched with bHLH/homeobox transcription factors that contain MYC binding motifs, raising the possibility that *MYC* may regulate expression of rescue genes.

Changes in cell state have been shown to be associated with resistance to anticancer therapies across multiple settings[45–48]. In the setting of BETi, epithelial-mesenchymal transition has been reported to be associated with resistance in pancreatic cells[49]. Among leukemias, stem cells have been found to be most resis-tant[50]—an opposite result to ours in medulloblastoma, where cells with a more (but not terminally) differentiated phenotype are less likely to respond to BETi.

Finally, our finding that BETi alters overall cell state towards a more differentiated neuronal phenotype is one that is highly relevant for medulloblastoma. The effect of BETi on the devel-oping brain remains to be characterized; such studies are essential to determine potential developmental sequelae of BETi when used to treat pediatric patients with medulloblastomas and other cancers.

## Methods

**Ethics statement**. Ethics approval was granted by relevant human IRB and/or animal research committees (IACUC) of Dana-Farber Cancer Institute (DFCI),

Boston Children's Hospital, The Broad Institute, Fred Hutchinson Cancer Research Center and Massachusetts Institute of Technology (MIT).

**Cell culture**. All cell lines included in this study have been shown to harbor features that recapitulate Group 3 human medulloblastomas[6,51]. D425 and D458 cells were a kind gift from Dr. Bigner. D283, CHLA01 and D341 cells were obtained directly from ATCC. MB002 cells were a kind gift from Dr. Cho[6]. HDMB003 cells were a kind gift from Drs. Pfister and Milde[52]. D425, D458, D283, and D341 cells were cultured in DMEM/F12 with 10% serum and 1% glutamate/pen-strep in ultra-low attachment flasks and plates. CHLA01 and MB002 cells were cultured in serum-free and growth factor supplemented media as previously described[6]. SNP-based fingerprinting assays were used prior to screens and sequencing assays to ensure authenticity. Mouse neural stem cells were generated and cultured as previously described[53]. All cells were routinely monitored for mycoplasma infection.

**Generation of drug-tolerant cells**. One million cells (D458 and D283) were passaged in 6-well plates in 1 μM JQ1 (or DMSO) control for greater than 30 days until they started to exhibit growth. DMSO treated cells were passaged every 3–4 days. Media was changed for the cells treated with the BET-bromodomain inhibitors every 3–4 days. Cells were counted weekly until BETi treated cells exhibited proliferation at which point cells were challenged with increasing doses of BETi (JQ1 and IBET151) to confirm the acquisition of drug tolerance.

**Generation of Cas9 expressing cells**. $1.5 \times 10^6$ cells with 4 μg/ml polybrene were seeded in one well of a 12-well plate, then spin-infected with pLX311-Cas9 virus. Cells were selected for 7 days in blasticidin (commencing 24 h infection). Cas9 activity was confirmed using eGFP reporter assays as previously described[54].

**Infection of pooled libraries: CRISPR-Cas9 screen and ORF screen**. Pooled lentiviral libraries were infected with a 30–50% infection efficiency, corresponding to a multiplicity of infection (MOI) of ~0.5–1. Spin-infections (2000 rpm for 2 h at 30 °C) were performed in 12-well plate format with $1.5 \times 10^6$ cells each well. Approximately 24 h after infection, cells were trypsinized and $3 \times 10^5$ cells from each infection, were seeded in two wells of a six-well plate, each with complete medium, one supplemented with the appropriate concentration of puromycin. Cells were counted four days post-selection to determine the infection efficiency. Volumes of virus that yielded ~30–50% infection efficiency were used for screening.

Screening-scale infections of the ORFeome pLX317 barcoded library (contains ~17,255 barcoded ORFs overexpressing 12,579 genes) and the Avana barcoded library (contains 73,687 barcoded sgRNAs targeting 18,454 genes and 1000 not-targeting guides). Cells were infected to achieve a representation of at least 1000 cells per ORF and 500 Cas9 expressing cells per CRISPR following puromycin selection. Cells within a replicate were harvested, pooled and split into T225 flasks 24 h after infection. Following selection, cells were seeded in T225 flasks in media. For the CRISPR-cas9 screens, cells were passaged for ~21 days. For the ORF rescue screens. 3 μM JQ1 and 5 μM of iBET-151 were added to the cells on Day 0. Cells were passaged in drug or fresh media containing drugs every 3–4 days. Cells were harvested ~21 days after initiation of treatment.

Each sgRNA vector and ORF vector harbor unique DNA barcodes that allow the tracking of abundance of each vector through the assays. Genomic gDNA extraction, PCR and sequencing were performed as previously described[55]. Samples were sequenced on a HiSeq2000 (Illumina).

CRISPR-Cas9: The processing of sgRNA read count data, quality control filters and modeling of guide activity, gene-knockout, and copy-number effect with the CERES algorithm were performed as previously described[54].

Determination of significance: 73,372 guides that passed quality control (including approximately 1000 guides that do not target any location in the reference genome as negative controls) were included in the analysis. To calculate the probability that a gene dependency score represents a true dependency in a given cell line, we fit a two-component mixture model in each cell line. The two components were (1) the empirically determined distribution of true dependent scores, identified using the pan-essential gene scores in that cell line, and (2) the empirically determined distribution of true non-dependent scores, identified as genes that were not expressed in that line. We defined as pan-essential genes 1607 genes whose dependency scores falls in the bottom 26% of gene scores in at least 90% of cell lines analyzed in the Achilles Avana dataset 18Q1, which includes 391 cell lines. The probability of dependency for each gene score is the probability that it was generated from the distribution of true dependent gene scores. To correct for noise in the tails of the distributions, all gene scores below $-1.5$ were assigned probability 1 of being dependencies and all gene scores above 0.25 were assigned probability 0. A Gaussian smoothing kernel with width 0.15 was applied to the final probability scores to further reduce noise. Genes with a dependency probability of >0.35 with a FDR < 0.2 were deemed to represent a dependency within each line.

ORF rescue screen: Read counts were normalized to reads per million and then $\log_2$ transformed. The $\log_2$ fold-change of each ORFs was determined relative to the initial time point for each biological replicate. Abundance of each ORF was measured at the initiation and completion of each assay to determine log-fold

changes following treatment with BET-bromodomain inhibitors. We defined "rescue ORFs" as those conferring >1.5 log-fold enrichment with a $q$ value of <0.25. We applied STRING[56] to identify direct and functional protein networks that exist between the entire set of candidate rescue ORFs identified across both cell lines.

**ORF rescue assays**. Medulloblastoma cells were transduced with lentivirus with pLEX-307 lentiviral vectors to overexpress eGFP, CCND2, CCND3, BCL2L1, MYOD1, MYOG, NEUROD1, NEUROG1, and NEUROG3 in individual infections. Cells were transduced using a spin protocol (3 million cells per infection, 2000 rpm for 2 h at 30 °C). Cells were harvested the following day and subjected to puromycin selection (1 μg/ml) at 48 h for three days. Cells were treated with 1 μM JQ1 or DMSO control. Cells were counted with trypan blue and cumulative doubling of JQ1 treated cells (relative to DMSO) between seven to 14 days. Genes that significantly attenuated BETi response (relative to eGFP controls) were deemed to "rescue" the BETi proliferative phenotype.

**Immunoblotting**. Cells were lysed in RIPA buffer containing protease and phosphatase inhibitors on ice for 60 min. Lysates for centrifuged at $13,000 \times g$ for 10 min and the supernatant was harvested. Supernatant was mixed with $4 \times$ SDS loading buffer and heated at 70 °C for 10 min and subjected to SDS-PAGE on 4–12% gradient gels. See Supplementary Data File 10 for antibodies used.

**Flow cytometry for apoptosis and cell-cycle**. Sensitive and drug-tolerant cells were treated with JQ1 (1 μM) or IBET151 (1 μM) for 72 h. Medulloblastoma cells overexpressing ORFs were selected in puromycin for 48 h before being treated with JQ1 (500 nM) or vehicle control for 72 h. Annexin V/Propidium iodide apoptosis assays were performed as previously described[6]. Proportion of cells in S phase was determined by flow cytometry assessment of BrdU/Propidium Iodine (BD biosciences) as per manufacturer's instructions.

**Whole-exome sequencing**. DNA was extracted from sensitive and drug-tolerant D425 and D458 cells (Qiagen DNeasy Blood and Tissue kit). DNA was subjected to whole-exome Illumina sequencing. Libraries with a 250 bp average insert size were prepared by Covaris sonication, followed by double-size selection (Agencourt AMPure XP beads) and ligation to specific barcoded adapters (Illumina TruSeq) for multiplexed analysis. Exome hybrid capture was performed with the Agilent Human All Exon v2 (44 Mb) bait set.

Sequence data were aligned to the hg19 (b37) reference genome with the Burrows-Wheeler Aligner (28) with parameters [-q 5 -l 32 -k 2 -t 4 -o 1]. Aligned data were sorted, duplicate-marked, and indexed with Picard tools. Base-quality score recalibration and local realignment around insertions and deletions was achieved with the Genome Analysis Toolkit.

Mutations were called from chronically passaged cells with MuTect, filtered against DNA from pretreatment samples, and annotated to genes with Oncotator.

**Gene-expression profiling**. Sensitive and drug-tolerant cells were treated with DMSO or JQ1 (1 μM) for 24 h in independent experiments (D283 = 3 replicates per condition, D458 = 5 replicates per condition) and RNA extracted (Qiagen RNeasy kit, with DNAse treatment, as per protocol). Gene expression profiles were assayed using Affymetrix Human Gene 2.0 ST microarrays. CEL files were RMA normalized[57]. Comparative marker selection analysis[58] was performed in Gene-Pattern using default settings. Genes with $p < 0.05$ and $q < 0.1$ were considered to have significant changes in expression unless otherwise specified. GSEA was performed using the C2 (CP) gene sets (MSigDB). Gene sets with nominal $q < 0.25$ were considered to be significantly altered. Principle component analysis was performed in R studio using the prcomp function.

**CyCIF staining of human tumors**. Formalin-fixed and paraffin-embedded (FFPE) tissue micro arrays (TMAs) containing medulloblastoma tissues from 50 patients and 10 normal brain controls were obtained from the Department of Pathology at Brigham and Women's Hospital according to IRB approval. TMAs were stained and analyzed on a single-cell basis using the open source t-CyCIF methodology[59] (http://www.cycif.org). The BOND RX Automated IHC/ISH Stainer (Leica Biosystems, Buffalo Grove, IL) was used to bake slides at 60 °C for 30 min, dewax with Bond Dewax Solution at 72 °C, and perform antigen retrieval with Epitope Retrieval 1 (ER1) solution at 100 °C for 20 minutes. TMAs underwent multiple cycles of antibody incubation, imaging, and fluorophore inactivation. Tissues were incubated overnight at 4 °C using commercially available, fluorophore-conjugated antibodies listed in Supplementary Data File 10. Nuclei were stained in each cycle using Hoechst 33342 (Catalog No. 4082S, Cell Signaling Technologies, Danvers, MA). Images were taken using the INCell Analyzer 6000 Cell Imaging System (GE Healthcare Life Sciences, Pittsburgh, PA) at 20× magnification using the DAPI, FITC, dsRed, and Cy5 channels. Exposure times ranged from 0.0750 to 1.000 ms. Fluorophores were inactivated by submerging slides in a 4.5% H2O2, 20 mM NaOH in PBS solution and exposed to LED light source for 2 h at room temperature.

Images were processed and single-cell data was quantified using customized versions of ImageJ and MATLAB scripts from the Sorger Lab GitHub repository

(https://github.com/sorgerlab/cycif). ImageJ was used to perform flat-field background subtraction, registration of images from each cycle, segmentation of single cells, and measurement of the average intensity value of each marker on a single cell basis. A cut off value for each marker was determined by visual inspection of the immunofluorescence image on ImageJ by a pathologist (S.C.). MATLAB was used to apply the cut off values, scoring cells with an average intensity value above the cut off as positive and cells with an average intensity value below the cut off as negative. All numeric analyses, including percentage of cells positive for one or more markers in each core of the TMA were performed in MATLAB (MathWorks Inc, Natick, MA).

**Global chromatin profiling**. Quantitative targeted mass spectrometry was performed on sensitive and drug-tolerant D425 and D458 cells passaged in DMSO or BETi (JQ1 or IBET151) for 24 h. Cell lysis, histone extraction and mass spectrometry (Global Chromatin Profiling) was performed as previously described[16]. Data was log normalized, and differentially altered marks were determined using comparative marker selection in GenePattern, correcting for cell-line, with a threshold of a Bonferroni FDR of <0.1.

**ChIP-sequencing**. Sensitive and drug-tolerant D425 and D458 were passaged in DMSO or JQ1 (1 μM) for 24 h. Sheared chromatin from each cell line was subjected to ChIP-sequencing as previously described[60], enriching for H3K27ac (Cell Signaling Technologies, 8173S)), H3K4me3 (Cell Signaling Technologies, 9751S), Histone H3 (Cell Signaling 4499S) and BRD4 (Bethyl, A301–985A100). ChIP libraries were indexed, pooled and sequenced on Illumina Hi-seq-2000 sequencers. Raw data was aligned to the human reference genome Hg19 using Picard tools (http://broadinstitute.github.io/picard/). Raw sequencing data was mapped to the reference genome using bowtie2 version 2.2.1 with parameters -p 4 -k 1. Peaks were called using MACS version 1.4.2 over an input control. Reads were extended 200-bp and normalized to read-density in units of reads per million mapped reads per bp (rpm/bp). To calculate ChIP-binding score for each gene, read-density in units for reads per million were aggregated for each gene (extending to 500 kb in each direction). Z-scores of ChIP-seq scores for each gene within each sample was calculated. Peaks and alignments were converted to TDFs by IGV tools and visualized by IGV.

**Drug assays**. JQ1, IBET151, and LEE011 assays were performed by seeding 1 million cells per well in 6-well ultralow attachment flasks with DMSO controls. Doses used (unless otherwise specified) were JQ1 1 μM, IBET151 1 μM, LEE011 500 nM or 1 μM. Total number of viable cells at designated time points were determined by trypan blue assays. dBET experiments were performed by seeding 1000 cells per well in 96-well plates (DMSO control or JQ1 2 μM for drug-tolerant cells) at the doses specified. Luminescence measurements of ATP content (Cell-Titer-Glo) were performed as a marker of cell viability.

**Loewe's synergy testing**. Experimental details: Cell lines were seeded into 384-well, white-walled, clear bottom plates at a density of 500 cells/well. Twenty-four hours after seeding, combination drugs were administered using an HP D300 Digital Dispenser in matrix format. Drug was administered such that the final volume of DMSO did not exceed 0.5%. The cells were then incubated for seven days and cell luminescence was measured using the Cell Titer-Glo assay (Promega, Madison, WI, USA) according to the manufacturer's instructions.

For calculation of drug combination effects: Curves were fit and deviation from the null model (Loewe additivity) assessed using the BIGL package in R (reference: PMID 29263342). General-purpose optimization (Nelder-Mean algorithm) was used for single-agent fits, and the "model" option was used to predict variance. Overall significance was assessed using the bootstrapped meanR test ($n = 1000$ iterations), and per-concentration significance using maxR, as described by the package authors.

**Animal studies**. In vivo studies were performed in compliance with IACUC approved protocols at Dana-Farber Cancer Institute (flank), Fred Hutchinson Cancer Research Center (Patient Derived Xenografts) or Massachusetts Institute of Technology (D458 intracranial experiments). Cells were tested for mycoplasma and subjected to IMPACT testing for pathogens prior to use in experiments.

**Flank**. Flank xenografts were established by injecting five million D458 cells or ten million MB002 cells (matrigel:PBS at a 1:1 ratio) in NSG mice (Jackson Labs). Mice were treated with JQ1 (50 mg/kg/daily intraperitoneal injection), LEE011 (75 mg/kg/daily oral gavage), combination therapy (LEE011 75 mg/kg/daily with JQ1 50 mg/kg/daily intraperitoneal injection) or vehicle control. Tumor growth was monitored by caliper measurements.

D458 Intracranial: D458 medulloblastoma cells were maintained in 1:1 DMEM/F12 media (Gibco) supplemented with 10% FBS (Hyclone), 1× Glutmax (Gibco), and 1× Pen/Strep (Sigma). Cells were transduced with a lentiviral *pLMP-GFP-Luc* vector to allow for stable expression of eGFP and firefly luciferase prior to implantation. Six-week-old NCR nude mice (Taconic) were used to generate intracranial orthotopic right cerebellar D458 medulloblastoma tumors.

In brief, mice were anesthetized using 2% isoflurane and their heads immobilized in a stereotactic headframe using atraumatic ear bars. A burr hole was made using a steel drill bit (Plastics One, Roanoke, VA, USA) 2 mm right of the sagittal and 2 mm posterior to the lambdoid suture. $10^5$ D458 cells were injected stereotactically into the right cerebellar hemisphere. Tumors were allowed to grow for 14 days prior to commencement of treatment (same doses as those used for flank injections). Intracranial tumor growth was monitored in vivo using bioluminescence IVIS® imaging (Xenogen, Almeda, CA) equipped with LivingImage™ software (Xenogen). Tumor response to treatment was tracked every 3–5 days using IVIS imaging. Mice were given 150 μL I.P. of 30 mg/mL D-luciferin (PerkinElmer) dissolved in PBS 10 min prior to IVIS imaging. Signal intensity was quantified within a region of interest using LivingImage™ software.

Intracranial patient derived xenografts: Med-114FH and Med-411FH models were implanted directly from the human patient into mouse cerebellum and propagated serially in mice for 5 (Med-114FH) or 6 (Med-411FH) passages as previously described[61]. To transduce with lentiviral mCherry-Luciferase, cells were briefly maintained in Neurocult media supplemented with their proprietary additive plus EGF and FGF before re-implantation in mice and continued serial passage in vivo. The cells used for this study were on mouse passage 4 (Med-114FH) or 7 (Med-411FH) after transduction. 100,000 cells per mouse were implanted orthotopically in the cerebellum of 5 (Med-114FH) or 7 (Med-411FH) week old HSD:Athymic Nude Foxn1nu #069 Envigo mice. Twenty-five days after implant mice underwent bioluminescent imaging for luciferase expression and were assigned treatment groups so as to normalize the luminescence across all groups, mice with luminescent signal <1e6 were excluded from the study, $n = 10$ per group. Mice were weighed and dosed daily with vehicle, LEE011 (75 mg/kg PO), JQ1 (50 mg/kg IP), or the combination. Hydrogel was provided as needed. Study endpoints included weight loss >20%, or observed morbidity such as mice being cold, hunched, or lethargic. The genomic characterization of these models as well as implant procedures have been previously reported[61].

**Short hairpin RNA (shRNA) suppression experiments**. D458 sensitive and BETi drug tolerant cells were transduced with lentivirus encoding shRNAs targeting BCL2L1, CCND2, or NEUROD1, in addition to SF3B1 (positive control) or eGFP (negative control). Cells were placed in puromycin selection 24 h after infection. On day 3 (48 h post-selection), 1000 cells were plated in each well of 96-well plates (five replicate wells per condition). Cell viability was measured on subsequent days by assessing ATP content with Cell Titre-Glo (Promega). Results were normalized to baseline.

**Barcoding**. 600K barcode library production: Five sets of primers were designed (Supplementary Data File 10) to incorporate a six nucleotide sub-pool barcode followed by 24 basepair degenerate sequence flanked by overhanging 5′AgeI site and 3′ EcoRI sites. Pairs of complementary pairs of oligos were annealed ligated into a modified pLKO.1 backbone. The modified pLKO.1 backbone had the human U6 promoter deleted (PpuMI-AgeI) and replaced with a short sequence (GGGGACCCAATGGACTATCATATGCTTACCGTAACTTGAAAGTATTT CGATTTCTTGGCTTTATATATCTTGTGGAAAGGACCGGT). This substitution will not transcribe the barcode sequence (https://portals.broadinstitute.org/gpp/public/resources/protocols). The ligations were amplified and plasmid preps generated and sequence verified for barcode diversity as previously described. Sub-pool 1 had a diversity of ~54,000 barcodes, while sub-pools 2–5 had a diversity of ~138,000–153,000 barcodes. (https://portals.broadinstitute.org/gpp/public/resources/protocols). Virus was generated and titered from each subpool, and mixed based on barcode diversity within the plasmids pools and titer levels to maximize a homogeneous distribution of representation of each barcode.

Cells were barcoded with the pooled lentiviral barcoding library. D283 and D458 cells were transduced with a low MOI (of 30%) with the aim of labeling single cells with individual barcodes. Each transduced cell line was expanded as single pools before being divided into replicate drug treatment (JQ1) or vehicle control experiments. We extracted DNA from each pool of cells to determine individual DNA barcode abundance prior to drug treatments. Cells were passaged in the presence of JQ1 (or DMSO control) for 40 days. Genomic DNA extraction, PCR and sequencing were performed as previously described[55] to determine the presence of each barcode (absolute and relative to the early time point control) in each replicate experiment. Barcodes with a minimum read count of 3 at the early time point were included for analysis.

**Statistical analysis**. Log-rank (Mantel-Cox) tests were performed to analze survival analysis of animal experiments. Unless otherwise described in the relevant results and methods, remaining $p$ values were determined using two-tailed $t$-tests. $p$ values of <0.05 were considered significant.

**Reporting summary**. Further information on research design is available in the Nature Research Reporting Summary linked to this article.

## Data availability

Gene expression profiling data has been deposited in GEO under accession number GSE122404. ChIP-sequencing data has been deposited in GEO under accession number GSE129521. Data from genome scale modifier screens and barcoding assays have been included in Supplementary Data Files and Data Source Files. Data from which figures were generated are included in Supplementary Data or Souce Data Files as indicated in individual figure legends. Uncropped western blots are included in the Data Source File.

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

## Acknowledgements

We gratefully thank and acknowledge the following funding sources for this project: Pediatric Brain Tumor Foundation (P.B., R.B.), St. Baldrick's Foundation (P.B.), Christopher Brandle Joy of Life Foundation (P.B., R.B.), Hannah's Heroes Foundation (P.B.), Alex's Lemonade Stand Foundation (R.B.), NIH K99 CA201592–02 (P.B.), NIH 1U54CA224068-01 (C.M.J.), the Jared Branfman Sunflowers for Life Fund for Pediatric Brain and Spinal Cancer Research (P.B., R.B., S.S.), Gray Matters Brain Cancer Foundation (R.B.), NIH R01CA188228 (R.B.), NIH R35 CA210030 (K.S.) and NIH R01 NS088355 (K.S.), NIH U54 CA225088 (SS and JL), The Bridge Project of MIT and Dana-Farber/Harvard Cancer Center (K.L.L. and R.B.), Stop and Shop Pediatric Brain Tumor Program (P.B., M.W.K.), NIH NCI U01 CA176058 (W.C.H.), NIH F32CA180653 (B.R. P.), the Damon Runyon Foundation (ABI is a Damon Runyon Foundation Fellow: DRSG:12–15) and The Isabel V. Marxuach Fund for Medulloblastoma Research (G.G. and P.B.), SS and JL were supported by U54 CA225088. We thank Dr. Alon Goren for technical advice and assistance with ChIP-seq assays, and Drs. David Feldman and Paul Blainey for assistance with analysis of barcoding data. We thank Mr. Eric Smith for his assistance in generating Fig. 1 (graphics). We gratefully acknowledge members of the Beroukhim, Johannessen, and Bandopadhayay labs, the Departments of Pediatric Oncology and Cancer Biology, Dana-Farber Cancer Institute and members of the Cancer Program, Broad Institute for their helpful discussions. Finally, we would like to thank and acknowledge the many children and families affected by pediatric brain tumors for their generous contributions to this research.

## Author contributions

P.B., C.J., and R.B. conceived the study. P.B., F.P., R.O., P.H., E.G.Z., G.B., K.Q., G.G., L. B., M.C., E.G., T.M., S.P., J.O., F.D., B.R.P., G.A., G.R., A.B., A.C., B.T., P.K., A.T., H.T., J. R.L., A.H., S.C., R.R., G.C., A.G., Y.L. K.S.C., C.S., F.C.L., M.P., M.Y., S.C., M.S., A.M., W. C.H., M.W.K., S.S., K.L.L., A.T., J.B., J.Q., P.C.G., J.J., D.R., F.V., K.S., C.J., and R.B. designed and/or executed experiments, or generated cell lines used in experiments. P.B., F.P., O.S., M.C., E.G., J.O., F.D., N.G., N.D., M.R., A.C., B.T., S.C., R.R., M.S., A.T., and R. B. contributed to data analysis. All authors contributed to the preparation of the manuscript. C.J. and R.B. supervised the overall study.

## Additional information

**Competing interests:** JEB is now an executive and shareholder of Novartis AG, and has been a founder and shareholder of SHAPE (acquired by Medivir), Acetylon (acquired by Celgene), Tensha (acquired by Roche), Syros, Regency and C4 Therapeutics. A.G., R.B., and K.S. also consulted for Novartis. P.B., K.S., and R.B. receive grant funding from Novartis. MWK is now an employee of Bristol Myer Squibb and C.M.J. is now an employee of Novartis. W.C.H. is a consultant for Thermo Fisher, AjuIB, Parexel, MPM and is a founder and advisor to K.S.Q. Therapeutics. SS consulted for Rarecyte, Inc. All other authors declare no competing interests.

Pratiti Bandopadhayay[1,2,3], Federica Piccioni[2], Ryan O'Rourke[1,2], Patricia Ho[1,2], Elizabeth M. Gonzalez[1,2], Graham Buchan[1,2], Kenin Qian[1,2], Gabrielle Gionet[1,2], Emily Girard[4], Margo Coxon[4], Matthew G. Rees[2], Lisa Brenan[2], Frank Dubois[2,5], Ofer Shapira[2,5], Noah F. Greenwald[2,5,6], Melanie Pages[1,2], Amanda Balboni Iniguez[1,2], Brenton R. Paolella[2,5], Alice Meng[7], Claire Sinai[1,7], Giovanni Roti[1,2,8], Neekesh V. Dharia[1,2,3], Amanda Creech[2], Benjamin Tanenbaum[2], Prasidda Khadka[1,2,3], Adam Tracy[2], Hong L. Tiv[9], Andrew L. Hong[1,2,3], Shannon Coy[10], Rumana Rashid[10,11], Jia-Ren Lin[12,13], Glenn S. Cowley[2,14], Fred C. Lam[15], Amy Goodale[2], Yenarae Lee[2], Kathleen Schoolcraft[7], Francisca Vazquez[2], William C. Hahn[2,7,16], Aviad Tsherniak[2], James E. Bradner[2,7,16,17], Michael B. Yaffe[2,15], Till Milde[18,19,20], Stefan M. Pfister[18,21,22], Jun Qi[5], Monica Schenone[2], Steven A. Carr[2], Keith L. Ligon[2,10,16,23,24], Mark W. Kieran[1,3], Sandro Santagata[7,10], James M. Olson[4], Prafulla C. Gokhale[9], Jacob D. Jaffe[2], David E. Root[2], Kimberly Stegmaier[1,2,3], Cory M. Johannessen[2] & Rameen Beroukhim[2,4,7,16]

[1]Dana-Farber/Boston Children's Cancer and Blood Disorders Center, Boston, USA. [2]Broad Institute of MIT and Harvard, Cambridge, USA. [3]Department of Pediatrics, Harvard Medical School, Boston, USA. [4]Clinical Research Division, Fred Hutchinson Cancer Research Center, Seattle,

USA. [5]Division of Cancer Biology, Dana-Farber Cancer Institute, Boston, USA. [6]Department of Neurosurgery, Brigham and Women's Hospital, Boston, USA. [7]Division of Medical Oncology, Dana-Farber Cancer Institute, Boston, USA. [8]Department of Medicine and Surgery, Hematology and BMT, University of Parma, Parma, Italy. [9]Experimental Therapeutics Core and Belfer Center for Applied Cancer Science, Boston, USA. [10]Department of Pathology, Brigham and Women's Hospital, Boston, USA. [11]Department of Biomedical Informatics, Harvard Medical School, Boston, USA. [12]Laboratory of Systems Pharmacology, Harvard Medical School, Boston, USA. [13]Ludwig Center for Cancer Research at Harvard, Harvard Medical School, Boston, USA. [14]Discovery Science, Janssen Research and Development (Johnson & Johnson), Spring House, PA, USA. [15]Koch Institute for Integrative Cancer Research, MIT, Cambridge, USA. [16]Department of Medicine, Harvard Medical School, Boston, USA. [17]Novartis Institutes for Biomedical Research, Basel, Switzerland. [18]Hopp Children's Cancer Center Heidelberg (KiTZ), Heidelberg, Germany. [19]CCU Pediatric Oncology, German Cancer Research Center (DKFZ), Heidelberg, Germany. [20]Department of Pediatric Oncology, Hematology, and Immunology, Center for Child and Adolescent Medicine, Heidelberg University Hospital, Heidelberg, Germany. [21]Division of Pediatric Neuro-Oncology, German Cancer Consortium (DKTK) and German Cancer Research Center (DKFZ), Heidelberg, Germany. [22]Department of Pediatric Hematology and Oncology, Heidelberg University Hospital, Heidelberg, Germany. [23]Department of Oncologic Pathology, Dana-Farber Cancer Institute, Boston, USA. [24]Department of Pathology, Boston Children's Hospital, Boston, USA

