## [Transparent Peer Review File · Nature Communications]

Reviewers' comments:

Reviewer #1 (Remarks to the Author):

In their manuscript, "Neuronal differentiation and cell-cycle programs mediate response to BET-bromodomain inhibition in MYC-driven medulloblastoma," Bandopadhyay et al. address the important question of mechanism of resistance to bromodomain inhibition. The bromodomain inhibitors are an important new focus in clinical trials for brain tumors and other cancers and the mechanisms of resistance and sensitivity are not yet described. The authors perform many elegant assays by which they identify candidate genes that mediate resistance to bromodomain inhibition. The data are well-presented and the manuscript is well-written. However, there are several major limitations to the work and relatively little that is novel with regard to the cancer biology and mechanisms of drug resistance. This significantly limits the impact of this work.

Major Comments

1. The study is very descriptive and does not provide a compelling case for how bHLH factors or cell cycle regulators are responsible for bromodomain inhibitor resistance. These are correlated findings only. It is not surprising that cells that maintain the ability to divide, exhibit expression and histone marks associated with expression in cell cycle regulatory genes. Neither is it surprising that there is discordant expression of transcription factors associated with differentiation in cells that remain competent for proliferation. This is a known phenomenon in cancer. The study would need to provide some insight into the mechanisms by which the expression of these genes is maintained in the setting of bromodomain inhibition.
2. The bulk of the data were generated in established cell lines and validated in a limited number of recently generated cell lines. This is problematic as the mechanisms of resistance to bromodomain inhibition may be a consequence of selection over years for growth under tissue culture conditions. More could be done with patient-derived specimens or murine specimens derived from published MYC-driven medulloblastoma models. It would have been stronger to nominate candidate genes in the primary models and use the cell lines to explore mechanisms.
3. The model is predicated on the resistant cells relying on the rescue genes identified in the ORF screen but this is not demonstrated. The authors would need to perform the CRISPR screen in the resistant cells to demonstrate that they required the rescue genes for resistance. It is possible that the cells adapt through an entirely independent mechanism and this would not have been evident in the current approach.
4. Statistics: There are several shortcomings in the statistics. First, the Western blot data as shown suggest they have been performed only once. There is no quantification and no indication of "n" and no statistics. This is not acceptable. There are experiments with only an "n" of 2 presented in Figure 4 with inappropriate statistical analysis. The experiment needs to be repeated at least three times for the valid statistical analysis. Multiple figure legends provide p values with no indication of what test was used. Finally, there is no statistics section in the method as required by Nature Communications.
5. The idea of predetermined resistance is really no different than phenotypic selection in the setting of any drug. There has not been adequate evaluation of phenotypic (transcriptional) diversity at the start. Single cell approaches to transcriptomic profiling would provide more convincing data but the concept is still not new.

Minor comments

1. Page 6: The authors say they did expression profiling of 5 lines but refer only to the main figure where we see only 2 lines. They should include a reference to Supplemental Figures 2 and 3
2. Page 9: The authors report 5 genes scored in both cell lines. Could this overlap be random?
3. Page 11: The first sentence should read (Figure 2E and Supplemental Figure 3).
4. Page 12: The statement, "Taken together, these data suggest that the drug-tolerant D458 cells exhibit attenuated phenotypic and transcriptional responses to treatment with BETi." Is an

overstatement. This would require transcriptome-wide analysis for support.

5. Page 19: The statement, "Our findings suggest a model whereby BETi suppresses essential bHLH lineage-specific transcription factors only in relatively undifferentiated cells, while more neuronally differentiated medulloblastoma cells maintain expression and viability in the presence of drug." Is a reach. These are transformed cells to speculate that aberrant co-expression of bHLH factors in cycling cells is a reflection of how the state of differentiation affects resistance is not supported.

Reviewer #2 (Remarks to the Author):

The manuscript by Brandopadhyay et al. describes mechanisms of BET inhibition resistance in cMYC amplified medulloblastomas and identifies potential combinatory drug therapies to overcome it. The authors used an elegant combination of gene expression profiling, genome-scale CRISPR/Cas9-mediated loss of function and ORF/cDNA driven rescue screens, and cell-based models of spontaneous resistance, to identify bHLH/homeobox transcription factors and cell-cycle regulators (CDK4/CDK6) as key mediators of the response to BET inhibitors. The experiments are comprehensive and convincing, particularly with respect to the therapeutic rationale for combining JQ1 with CDK inhibitors in a deadly MB subgroup. The link made between JQ1 treatment and neuronal differentiation is somehow the weakest part of this work as it is not clear how cells that are in an 'intermediate' differentiation state between stem cells and neurons can continue to exist, proliferate and populate JQ1-treated tumors.

There are mostly minor comments that would strengthen this work if addressed:

1. Page 11: "our analysis here confirmed MYC to be an essential gene that is transcriptionally suppressed by BETi in both D458 and D282... is not the sole mediator of BETi's phenotypic effects"

a. Which analysis/figure supports this claim?

b. Why wasn't MYC included in ORF screens? It would be important to understand the relative resistance to BETi conferred by MYC as compared to other rescue genes (cell cycle regulators, BCL2, bHLH) and the consistency of rescue across medulloblastoma cell lines since MYC may be the primary mediator of resistance to BETi through indirect effects on transcription of the rescue genes

2. Supplementary Fig. 4B/C: it appears that there is more apoptosis and necrosis in drug-tolerant cells at baseline (in DMSO) – is this relationship statistically significant and if so, how does increased apoptosis and necrosis in drug-tolerant cells align with the model of drug-tolerant cells exhibiting 'intermediate' cell cycle frequency between stem-like cells and terminally differentiated neurons?

3. Supplementary Fig. 4: Figure legend should be revised (and reference in text) as the naming of F-G is different from the figure legend.

4. Fig. 3C/D: The authors used a previously-described JQ1 transcriptional signature to classify drug-sensitive and tolerant cells. What were the differentially expressed genes between the JQ1-sensitive and tolerant cells using a naïve, unbiased approach, and how many of these genes overlapped with the previously-described JQ1 signature?

5. Fig. 3E/F: Quantitation should be provided for immunoblots

a. It is not evident that BCL2L2, CCND2 and HLX are suppressed by JQ1 in drug-sensitive cells, as claimed in the text

6. Fig. 4A: Because BET inhibitors target K27ac, it would be good to include the K27ac profile by mass spectrometry, even if this histone mark is not differentially altered in drug-tolerant lines. Specific peptides with multiple PTM combinations are differentially found between sensitive and drug tolerant states and authors describe an overall trend towards activating PTMs specific to tolerant state and repressive PTMs in sensitive state. What is the overall fold change in sum abundance of marks such as H3K27me3 and H3K9me2/3?

7. Page 14: "tolerant cells exhibited increased levels of the non-repressive mono and di-methyl forms of H3K27"
- a. The functions of H3K27me1/2 are not well characterized and it is not clear that these are non-repressive marks
8. Fig. 4B/C: What were the changes in H3K4me3 and H3K27ac ChIP-binding at baseline (DMSO) in drug-resistant cells? All other graphs have this comparison
9. Fig. 4D: It is not clear based on ChIP tracks that there is increased H3K27ac at the promoter or enhancer of NEUROG1 in drug-tolerant cells
10. Supplementary Fig. 6A: Quantitation of immunoblot is necessary
 - a. Why is TUJ1 band of a different size in NEUROG1-expressing cells?
11. Fig. 5A: Format of immunoblot is misleading: the DMSO control should be on the left and JQ1-treated to the right in drug-tolerant cells, as was done throughout the paper
12. Page 16: "The drug-tolerant cells also had increased expression of neuronal markers NEUROG1, NEUROD1, TUJ1 and NF68"
Which figure shows increased expression of NEUROG1, NEUROD1 and TUJ1 in drug-tolerant cells?
13. ChIP normalization by spike in would benefit the quantitative description of changes in H3K27ac over targets described to be implicated in JQ1 resistance.

14. Does relaxation of stringency thresholds (p value range) of gene expression data and Rescue genes from ORF screen increase the number of overlapping hits? Where is CCND2 ranked in overall datasets for D238 line? Do you find more members of pathways of interest (BCL2 family, developmental genes) when thresholds are relaxed?

Reviewer #3 (Remarks to the Author):

This manuscript focuses on the mechanisms of resistance to BET bromodomain inhibitors (BETi) in medulloblastoma (MB). Using a combination of RNAseq to identify genes that are downregulated by BETi, a CRISPR screen to discover genes that are essential for cell survival, and an ORF overexpression screen to find genes that can rescue cells from the effects of BETi, they come up with a list of genes that are likely to be key mediators of BET inhibition. These genes include regulators of the cell cycle such as D-cyclins, regulators of apoptosis such as BCL2 family members, and bHLH transcription factors (including members of the NeuroD and NeuroG families) that regulate cell fate and differentiation. They show that several of these genes are upregulated in cells that have become resistant to BET inhibitors, and that this upregulation is a consequence of epigenetic reprogramming. Finally, they demonstrate that counteracting the upregulation of cyclins by treating cells with CDK4/6 inhibitors can overcome resistance and synergize with BET inhibitors to antagonize tumor growth. Based on these studies, they propose that combination therapy with BET inhibitors and CDK4/6 inhibitors will be effective for MYC-driven MB.

This paper is a technical tour-de-force, combining an array of cutting edge functional genomic approaches to elucidate the mechanisms of BETi activity. Overall, the experiments are well designed and controlled, and the conclusions are well supported by the data. My main concern is that many of the experiments are not described with sufficient clarity and in sufficient detail to be comprehensible to the average reader. There are also some instances where the figures are not clearly presented, or where the data are not presented in figures at all. Most of these issues should be addressable with revisions to the text and figures, and with a few fairly straightforward experiments. Details are presented below:

1. Most of these studies were done on cell lines that have been in culture for several decades. It is understandable that some of the approaches – eg. CRISPR screening, ORF screening – would be challenging to carry out using PDXs or mouse models, but the authors should at least discuss the limitations of the cells they are using, and if possible, perform some of the validation studies on

patient-derived xenografts that more closely resemble the human disease.

2. The authors state (p. 6) that they performed expression profiling on 5 MYC-driven MB lines, but in Supplemental Table 1 they only present data on 4 lines. In addition, few details are given about this experiment: what concentration of BETi was used, and how long were cells treated with BETi before gene expression was analyzed? Some of this information is in the Methods section, but it really should be spelled out in the text and/or legends so readers can follow what was done. Also, it is not clear how effective JQ1 was at inhibiting growth/survival in the cells used here. Even if similar data were shown in a previous paper, it would be helpful to show growth or survival curves here so that readers can appreciate the effects for which a mechanistic explanation is being sought. Finally, the gene expression data are discussed very briefly, and not presented anywhere except in Supplementary Table 1. It would be helpful to see some kind of graphical representation of these data, e.g. a heat map, a cytoscape plot of the pathways that were altered, etc.

3. The CRISPR screen is also poorly described in the text. How were cells cultured, and for how long? What assay was used to determine dependency, and what were the criteria for a gene to be considered a dependency? Again, this information might be in the methods, but it is essential to provide at least some detail in the text and in the figure/table legends so that readers can follow the experiment. Finally, the authors state (p. 7) that the dependencies were enriched for members of 39 pathways, but no information is provided about how these pathways were identified. The pathways are shown in Supp. Table 3, but the genes in these pathways are not listed, which makes it difficult to evaluate their significance. This information should be added to the table. Some graphical representation of the CRISPR screen results would also be helpful.

4. In describing the ORF screen, the authors state that they “applied a lentivirally delivered ORF library...” in cell lines treated with BETi or vehicle, but they do not describe anything else about the assay or the criteria for determining whether a gene rescued cells from BETi. They state that “Log-fold-changes for each gene were highly correlated...”, but it is not clear what was being measured (gene expression?) and what was being compared to what to determine fold change. These are not simple or standard assays, and the reader needs to be brought along for the ride, rather than being asked to simply accept the conclusions.

5. The downstream analysis of the ORF screen results is also not very clearly explained. “Proteins related to cell fate commitment, transcription and developmental processes were significantly enriched in the rescue gene network”: what is the rescue gene network and how was it derived? “This pathway enrichment was reflected by a high connectivity for the ORF network as a whole, with 31 nodes and 42 edges” What does high connectivity mean? What are nodes and edges?

6. The low-throughput rescue assays should also be clarified. What assay was used to determine rescue, and what were the criteria for a gene to be considered to have rescued cells from JQ1? Also, in describing the results of these assays, the authors mention CCND2 and CCND3 rescuing D458 cells, but they do not mention BCL2L1, which appears to be significant based on Figure 2E.

7. On the top of p. 11, they refer to Supplemental Figure 2E, but there is no panel E in Supp. Fig. 2.

8. The authors state (p. 11) that they sought to determine if rescue genes were differentially expressed in “naturally arising” models of BETi resistance. The term naturally arising is somewhat misleading: treating a long-term passaged cell line with an inhibitor for a month is not at all natural. The authors should use another phrase to describe these experiments.

9. In discussing Figure 3 the authors state that they probed expression profiles of BETi sensitive and tolerant cells for previously described signatures of JQ1 transcriptional response. The genes they used for these signatures should be included in a Supplemental Table. They also state that expression of JQ1 signature genes increased following removal of JQ1, but they don't cite a figure

or table in which these data are presented.

10. On p. 13, they state that expression of NEUROD1 and NEUROG1 “also increased with treatment of BETi”. This seems to contradict previous statements that these genes decreased following BETi treatment. Do the authors mean that these genes decrease in cells that are resistant to BETi?

11. On p. 16, the authors state that “The drug-tolerant cells also had increased expression of neuronal markers NEUROG1, NEUROD1, TUJ1 and NF68, and suppression of the stem marker MSI1.” The only Figure cited in this section is Figure 5, and while this figure shows data for NF68 and MSI1, there are no data for NEUROG1, NEUROD1 or TUJ1. Are those genes presented elsewhere?

12. The authors hypothesize that resistance to BETi is associated with acquisition of a cellular state that is primed for differentiation, and later show that resistance to BETi is predetermined. These concepts – that the resistant state is acquired and that it is pre-existent – are quite different. The authors should discuss this more extensively and try to reconcile them into a single coherent hypothesis. Moreover, their description of the more differentiated resistant cells sounds a lot like what some investigators refer to as “transit amplifying cells”. The authors should explore this concept in their discussion.

13. After an extraordinary amount of functional genomics and epigenomics, the authors identify bHLH transcription factors, BCL2 family members and D-Cyclins as key mediators of BETi and of resistance to these drugs. Among these, they focus on D-Cyclins, and test the efficacy of the CDK4/6 inhibitor LEE011 (ribociclib) as a strategy for overcoming BETi resistance. Although ribociclib has not been extensively studied in MB, the related agent palbociclib has. The authors should cite the previous papers on palbociclib (Cook Sangar et al., Clin Cancer Res 2017; Hanaford et al., Clin Cancer Res 2016; Whiteway et al., J. Neurooncol 2013) and discuss their results in the context of those papers.

14. It is unfortunate that the authors did not test the functional importance of the other two classes of genes they identified: bHLH transcription factors and BCL2 family members. Although there are not specific drugs that can be used to inhibit bHLH transcription factors, knocking them down and testing the effects on resistance to BETi would enhance the novelty and significance of these studies. Similar knockdowns, or pharmacological inhibitors such as Navitoclax or Venetoclax or APG-1252, could also be informative regarding the importance of BCL2 family members.

15. The authors state that they observed synergy between JQ1 and LEE011, but based on Supp. Fig. 6 they only tested two concentrations of LEE011 and one concentration of JQ1. It is unclear how they selected these particular concentrations, and what would have happened at other concentrations. The authors should use a more rigorous approach to test synergy, with multiple concentrations of each drug, such as the approach described in Chou et al., Cancer Res. 2010.

16. Although the combination of JQ1 and LEE011 is more effective than either drug alone, Figure 5G shows that mice treated with the combination still succumb to their disease. The authors should discuss why this might be happening. When tumors are harvested from dually-treated animals, are they now resistant to both drugs? If so, what strategies could be used to overcome this resistance? Would inhibition of BCL2L1 or one of the bHLH proteins help abolish resistance?

Reviewer #1 (Remarks to the Author):

In their manuscript, "Neuronal differentiation and cell-cycle programs mediate response to BET-bromodomain inhibition in MYC-driven medulloblastoma," Bandopadhyay et al. address the important question of mechanism of resistance to bromodomain inhibition. The bromodomain inhibitors are an important new focus in clinical trials for brain tumors and other cancers and the mechanisms of resistance and sensitivity are not yet described. The authors perform many elegant assays by which they identify candidate genes that mediate resistance to bromodomain inhibition. The data are well-presented and the manuscript is well-written. However, there are several major limitations to the work and relatively little that is novel with regard to the cancer biology and mechanisms of drug resistance. This significantly limits the impact of this work.

Major Comments

1. The study is very descriptive and does not provide a compelling case for how bHLH factors or cell cycle regulators are responsible for bromodomain inhibitor resistance. These are correlated findings only. It is not surprising that cells that maintain the ability to divide, exhibit expression and histone marks associated with expression in cell cycle regulatory genes. Neither is it surprising that there is discordant expression of transcription factors associated with differentiation in cells that remain competent for proliferation. This is a known phenomenon in cancer. The study would need to provide some insight into the mechanisms by which the expression of these genes is maintained in the setting of bromodomain inhibition.

We thank the Reviewer for the comment that "The authors perform many elegant assays by which they identify candidate genes that mediate resistance to bromodomain inhibition"

We reiterate that we applied multiple independent genomic approaches to test the susceptibility of medulloblastoma models to genes that we found to be downregulated by BRD4 inhibition. We validated these findings in numerous medulloblastoma models and evaluated mechanisms through which these genes are reexpressed in drug tolerance. We note that Reviewer 3 also stated, "This paper is a technical tour-de-force, combining an array of cutting edge functional genomic approaches to elucidate the mechanisms of BETi activity. Overall, the experiments are well designed and controlled, and the conclusions are well supported by the data."

We also highlight the following novel findings:

- 1. The mediators of BETi response have not previously been systematically characterized. Through our integrated genomics approach, we identified three key pathways (BCL2 family members, cell-cycle regulators and bHLH and homeobox transcription factors) as all mediating the phenotypic effects of BETi.*
- 2. To validate these genes as mediators of response, we performed over-expression rescue experiments to demonstrate that they are able to partially rescue from the phenotypic effects of BETi. Furthermore, we find that these pathways are re-expressed in cells that have acquired resistance to BETi.*
- 3. We also respectfully disagree with the Reviewer that these findings are not surprising. While we agree that cells that maintain proliferative advantage express cells that induce cell-cycling, our paper is focusing on cells that re-institute their ability to proliferate despite concurrent treatment with BETi, through the re-expression of genes that mediate BETi sensitivity.*
- 4. We show that expression of these genes is mediated through changes in chromatin structure, with the redistribution of activating marks preferentially to genes that mediate BETi resistance.*

2. The bulk of the data were generated in established cell lines and validated in a limited number of recently generated cell lines. This is problematic as the mechanisms of resistance to bromodomain inhibition may be a consequence of selection over years for growth under tissue culture conditions. More could be done with patient-derived specimens or murine specimens derived from published MYC-driven medulloblastoma models.

It would have been stronger to nominate candidate genes in the primary models and use the cell lines to explore mechanisms.

Our study included five distinct MYC-driven medulloblastoma cell lines that have been shown to recapitulate the human disease(1). This included the recently generated MB002 cell line, which was established in 2012 by Dr. Cho at Stanford(2). This line is not passaged in serum and has been shown to recapitulate Group 3 medulloblastomas(2).

Our revised manuscript includes validation in a further three lines, including in vitro combination testing of a model generated from our collaborators in Heidelberg(3), and in vivo testing of two patient derived xenograft models generated by our collaborator Dr. Olson at the Fred Hutchinson Cancer Research Center(4).

3. The model is predicated on the resistant cells relying on the rescue genes identified in the ORF screen but this is not demonstrated. The authors would need to perform the CRISPR screen in the resistant cells to demonstrate that they required the rescue genes for resistance. It is possible that the cells adapt through an entirely independent mechanism and this would not have been evident in the current approach.

We agree with the Reviewer that suppressing the rescue genes from the ORF screen in the resistant cells would strengthen our manuscript. We therefore suppressed expression of the rescue genes CCND2, BCL2L1 and NEUROD1 in both sensitive and drug-tolerant D458 cells, confirming ongoing dependence on these genes in the drug-tolerant setting. We included these data in the revised manuscript as described below:

‘To further evaluate the functional significance of the anti-apoptosis, cell cycle and bHLH transcription factors in drug-tolerance, we suppressed BCL2L1, CCND2 and NEUROD1 using short hairpin RNAs, confirming each of these genes to be essential in both D458 drug-naïve and -tolerant cells (p values <0.0001 in all cases, Figures 3G, 3H and Supplemental Figures 11C-11D). We concluded that rescue genes from cell-cycle, bHLH transcription factor, and anti-apoptotic pathways are cell-essential in D458 cells and are re-expressed in cells that acquire drug-tolerance, remaining genetic dependencies in those cells’.

4. Statistics: There are several shortcomings in the statistics. First, the Western blot data as shown suggest they have been performed only once. There is no quantification and no indication of “n” and no statistics. This is not acceptable. There are experiments with only an “n” of 2 presented in Figure 4 with inappropriate statistical analysis. The experiment needs to be repeated at least three times for the valid statistical analysis. Multiple figure legends provide p values with no indication of what test was used. Finally, there is no statistics section in the method as required by Nature Communications.

We clarify that all experiments (including Western blots) had been performed in at least three replicate experiments. We now include densitometry measurements across western blots showing differences with p values as requested. The original Figure 4 depicted the difference between drug naïve cells and drug tolerant cells (with two drug naïve cell replicates in DMSO and two in JQ1). At the request of the Reviewer, we now include a minimum of three replicates for our ChIP-seq analysis in all conditions. We have included the specific test to determine p values in all figure legends and included a dedicated statistics section in the methods.

5. The idea of predetermined resistance is really no different than phenotypic selection in the setting of any drug. There has not been adequate evaluation of phenotypic (transcriptional) diversity at the start. Single cell approaches to transcriptomic profiling would be provide more convincing data but the concept is still not new.

We agree that profiling of the heterogeneity of pre-treatment medulloblastomas is interesting. Our manuscript includes multiplexed profiling of differentiation markers across a panel of 46 human medulloblastomas (Figure 6C and Supplemental Table 9). These data revealed medulloblastomas to be heterogeneous tumors, with subpopulations of cells exhibiting stem markers, some exhibiting more differentiated markers, and other populations to exhibit mixed phenotypes.

We also agree that single cell RNA-sequencing approaches of primary human tumors are helpful in profiling transcriptional heterogeneity. Indeed, a pre-print reporting the single-cell transcriptional profiles of pediatric cerebellar tumors including medulloblastoma, has been deposited to BioRxiv(5). This study reports Group 3 medulloblastomas to exhibit transcriptional diversity including stem cell populations in addition to more differentiated populations that arise from multiple lineages.

We have updated our Discussion to refer to these data:

‘Brain tumors are comprised of a hierarchy of subpopulations of cells, ranging from those that resemble stem cells with self-renewal capacity to those in which lineage commitment has been triggered through the expression of transcription factors(6-8). Our analysis of stem and lineage markers support similar heterogeneity within (and between) medulloblastomas, a finding that has also been reported in single cell RNA-sequencing profiles of human medulloblastomas(5). We identified cells that express both genes associated with neuronal differentiation and stemness as being associated with drug-tolerance, and found similar cells to be present in both pretreatment medulloblastoma cell lines and primary tumors, raising the possibility that selection of these intrinsically drug-tolerant cells contribute to BETi tolerance. Similar populations of cells have been identified in other normal developmental hierarchies, and in cancer, and have been termed transit amplifying cells(9, 10) or in the neural context, ‘activated quiescent neural progenitor cells’. These cells have been shown to harbor a phenotype that is more differentiated than stem cells, and to harbor increased proliferate potential compared to quiescent stem cells or terminally differentiated neurons.’

Minor comments

1. Page 6: The authors say they did expression profiling of 5 lines but refer only to the main figure where we see only 2 lines. They should include a reference to Supplemental Figures 2 and 3

We apologize for this confusion. We were referring to the gene expression profiling that we had performed across five MYC-driven lines and published in our CCR paper (Bandopadhyay et al, 2014). However, we have also performed additional profiling of four lines which form the basis for the gene-expression analysis presented in this manuscript. We have amended text to refer to expression profiling to these four MYC-driven lines and cite Supplemental Table 1 and Supplemental Figures 1 and 2 as below:

‘To characterize the extent and uniformity of transcriptional effects of BETi in medulloblastoma models, we performed expression profiling of four MYC-driven medulloblastoma cell lines treated with the BET-bromodomain inhibitor JQ1, relative to vehicle controls (Supplemental Table 1 and Supplemental Figures 1 and 2).’

2. Page 9: The authors report 5 genes scored in both cell lines. Could this overlap be random?

We agree with the Reviewer that it is important to test the statistical significance of this overlap, and had therefore included the p value of 0.0001, which was computed using a Chi-Square test. We have now edited this sentence to make it clearer that this overlap was statistically significant, and therefore unlikely to be random.

‘First, five of these rescue ORFs (*ATOH1*, *BCL2L1*, *BCL2L2*, *CCND3* and *NEUROG1*) were common to both cell lines, a statistically significant overlap ($p < 0.0001$).’

3. Page 11: The first sentence should read (Figure 2E and Supplemental Figure 3).

Thank you for noting the error in citation of the Figures. We have corrected this to refer to Supplemental Figures 6C and Supplemental Figure 7, which include the additional validation of lines MB002, D341 and CHLA01.

4. Page 12: The statement, “Taken together, these data suggest that the drug-tolerant D458 cells exhibit attenuated phenotypic and transcriptional responses to treatment with BETi.” Is an overstatement. This would require transcriptome-wide analysis for support.

We agree that a transcriptome-wide analysis of differentially expressed genes in drug-sensitive and drug-tolerant cells would further support our conclusion that drug-tolerant cells exhibit attenuated transcriptional responses to BETi. We have included the following analysis to the manuscript:

‘We hypothesized that drug-tolerant cells evade BETi effects by reversing its transcriptional consequences. We performed genome-scale expression profiling of sensitive and drug-tolerant cells following treatment with DMSO or JQ1. We found 3,279 genes to be significantly upregulated in drug-tolerant cells cultured in JQ1 compared to drug-naïve cells (Supplemental Figure 8D, Supplemental Table 5A). These were significantly enriched for transcriptional targets of JQ1 in D458 cells (1667 genes, p value <0.0001). They were also enriched for genes that have previously been found to be suppressed by BETi both across cancers (JQ1 consensus signature(2, 11) Supplemental Table 5B, Figure 3C) and among medulloblastoma cell lines(2) (Figure 3D) (p value <0.0001 in both cases). However, removal of JQ1 increased expression of both of these genesets in drug-tolerant cells even further (Figures 3C and 3D, p<0.0001 for both genesets), suggesting residual JQ1 activity in these cells. These cells also exhibited increased expression of BET-bromodomain target pathways including MYC activation and E2F signaling(2, 11, 12) and of MYC itself (Supplemental Figure 9A and 9B). Taken together, these data suggest that the drug-tolerant D458 cells exhibit attenuated phenotypic and transcriptional responses to treatment with BETi.’

5. Page 19: The statement, “Our findings suggest a model whereby BETi suppresses essential bHLH lineage-specific transcription factors only in relatively undifferentiated cells, while more neuronally differentiated medulloblastoma cells maintain expression and viability in the presence of drug.” Is a reach. These are transformed cells to speculate that aberrant co-expression of bHLH factors in cycling cells is a reflection of how the state of differentiation affects resistance is not supported.

We include data in our manuscript that:

1. Validates bHLH/homeobox transcription factors to be rescue genes for BETi in medullo
2. Confirmed that expression of these transcription factors drive primary neural stem cells towards a neuronal phenotype.

We now included discussion in the manuscript regarding how these findings may related to transit amplifying cells as suggested by Reviewer 3 (and see major comment five above).

Reviewer #2 (Remarks to the Author):

The manuscript by Bandopadhyay et al. describes mechanisms of BET inhibition resistance in cMYC amplified medulloblastomas and identifies potential combinatory drug therapies to overcome it. The authors used an elegant combination of gene expression profiling, genome-scale CRISPR/Cas9-mediated loss of function and ORF/cDNA driven rescue screens, and cell-based models of spontaneous resistance, to identify bHLH/homeobox transcription factors and cell-cycle regulators (CDK4/CDK6) as key mediators of the response to BET inhibitors. The experiments are comprehensive and convincing, particularly with respect to the therapeutic rationale for combining JQ1 with CDK inhibitors in a deadly MB subgroup. The link made between JQ1 treatment and neuronal differentiation is somehow the weakest part of this work as it is not clear how cells that are in an 'intermediate' differentiation state between stem cells and neurons can continue to exist, proliferate and populate JQ1-treated tumors.

We thank Reviewer 2 for their comments. We agree that it is important to demonstrate that populations of cells that exhibit an intermediate differentiation state exists in medulloblastoma. To this end, our manuscript includes:

1. CyclF profiling for a cohort of primary human medulloblastomas that demonstrate subpopulations of cells that harbor both neuronal and stem markers
2. Profiling of drug-tolerant cells that reveal cells to express both neuronal and stem markers
3. Cell-cycle analysis that confirms presence of BrdU positive drug-tolerant cells (as a marker of S-phase)

In addition, Reviewer 3 raised the intriguing possibility that these cells are similar to 'transit amplifying cells' that have been described in both normal developmental hierarchies and in cancer, and which exhibit the phenotypes that we observe: moderate differentiation with continued proliferation. We have added the following text to our Discussion:

'Brain tumors are comprised of a hierarchy of subpopulations of cells, ranging from those that resemble stem cells with self-renewal capacity to those in which lineage commitment has been triggered through the expression of transcription factors(6-8). Our analysis of stem and lineage markers support similar heterogeneity within (and between) medulloblastomas, a finding that has also been reported in single cell RNA-sequencing profiles of human medulloblastomas(5). We identified cells that express both genes associated with neuronal differentiation and stemness as being associated with drug-tolerance, and found similar cells to be present in both pretreatment medulloblastoma cell lines and primary tumors, raising the possibility that selection of these intrinsically drug-tolerant cells contribute to BETi tolerance. Similar populations of cells have been identified in other normal developmental hierarchies, and in cancer, and have been termed transit amplifying cells(9, 10) or in the neural context, 'activated quiescent neural progenitor cells'. These cells have been shown to harbor a phenotype that is more differentiated than stem cells, and to harbor increased proliferate potential compared to quiescent stem cells or terminally differentiated neurons.'

There are mostly minor comments that would strengthen this work if addressed:

1. Page 11: "our analysis here confirmed MYC to be an essential gene that is transcriptionally suppressed by BETi in both D458 and D282... is not the sole mediator of BETi's phenotypic effects"
a. Which analysis/figure supports this claim?

MYC scored as a dependency in the CRISPR-cas9 screens and its mRNA expression is significantly suppressed with treatment with BETi. These results are in the supplemental tables 1 and 2 which we have cited in the text below.

'BETi has been reported as a means to target MYC(2, 12). MYC was not included in the ORF screens. However, we previously demonstrated that ectopic MYC expression rescues D283 cells from BETi(2), and our analysis here confirmed MYC to be an essential gene (Supplemental Table 2) that is transcriptionally suppressed by BETi in both D458 and D283 (Supplemental Table 1)—indicating that MYC also fulfills all three criteria of a key essential gene that is suppressed by BETi. However, our analysis indicates that MYC is not the sole mediator of BETi's phenotypic effects.'

However, our analysis of cell essential genes that are suppressed by BETi indicated that MYC was one of many cell-essential genes that are suppressed by BETi. Figure 2 demonstrates 876 such genes in D458 and 760 genes in D283, while similar analysis revealed 504 essential genes to be suppressed by BETi in D341, and 1005 genes in D425 (shown in Supplemental Figure 3).

- b. Why wasn't MYC included in ORF screens? It would be important to understand the relative resistance to BETi conferred by MYC as compared to other rescue genes (cell cycle regulators, BCL2, bHLH) and the consistency of rescue across medulloblastoma cell lines since MYC may be the primary mediator of resistance to BETi through indirect effects on transcription of the rescue genes

MYC is not included in our current ORF library due to technical difficulties in expressing it. However, we have attempted to express MYC in separate experiments focused on this gene.

In these MYC-focused experiments, we were unable to induce stable ectopic expression of *MYC* in these *MYC*-driven medulloblastoma cell lines. We hypothesize that high levels of *MYC* are toxic to endogenously high *MYC* expressers.

We have demonstrated that MYC was able to partially rescue cells from JQ1 in D283 cell lines, and included these experiments in a prior publication which we cite in the manuscript (CCR, 2014). To accomplish these experiments, we commenced BETi within hours of expressing MYC. This apparently overcame MYC-related toxicities, possibly because BETi suppressed MYC expression.

We agree with the Reviewer that MYC may still play an important role in drug-tolerant cells. Indeed, we have found MYC to be reexpressed in drug tolerant cells (Supplemental Figure 11A), and for the Rescue ORFs to be enriched with bHLH transcription factors that harbor MYC binding motifs (Supplemental Figure 3F). We have highlighted this enrichment in the Results, and also added text to the Discussion to highlight the potential role of MYC in mediating drug tolerances.

Results:

‘Proteins related to cell fate commitment, transcription, and developmental processes were significantly enriched in the rescue gene network ($q < 0.0001$), as were MYC-type basic helix-loop-helix (bHLH) ($q < 0.001$), cyclin ($q < 0.01$) and myogenic basic muscle-specific protein domains ($q < 0.001$, Supplemental Figure 3F). Protein network analysis (performed using String, see Methods) revealed that the pathway enrichment was also reflected by a high connectivity for the ORF network as a whole, 42 edges (edges refer to protein-protein interactions) expected number of edges: 8) with 31 nodes (individual ORFs) and a clustering coefficient of 0.749 ($p < 0.0001$, Figure 2C).’

Discussion:

‘We have also identified bHLH/homeobox transcription factors and BCL2 family members as mediating response and resistance to BETi. Future work will examine whether also inhibiting these pathways (in particular BCL2 family members) may represent a therapeutic strategy to attenuate resistance to the combination of cell-cycle inhibition and BETi. While our integrative analysis has highlighted multiple mediators of BETi response, *MYC* remains an important target. We have previously found *MYC* to be a mediator of BETi response in *MYC*-driven medulloblastoma(2). In our current study, we have found *MYC* to be re-expressed in drug-tolerant cells, and for rescue ORFs to be enriched with bHLH/homeobox transcription factors that contain MYC binding motifs, raising the possibility that *MYC* may contribute to expression of genes that contribute to the development of drug tolerance of BETi.’

2. Supplementary Fig. 4B/C: it appears that there is more apoptosis and necrosis in drug-tolerant cells at baseline (in DMSO) – is this relationship statistically significant and if so, how does increased apoptosis and necrosis in drug-tolerant cells align with the model of drug-tolerant cells exhibiting ‘intermediate’ cell cycle frequency between stem-like cells and terminally differentiated neurons?

Thank you for this interesting observation. First, we should clarify that the drug-tolerant cells are always maintained in JQ1. Therefore, ‘baseline’ for these cells represents culture conditions that contain JQ1. When we compare the drug-tolerant cells in JQ1 to the drug-sensitive cells (also in JQ1), we observe a significant reduction in the percentage of necrotic drug-tolerant cells (p value 0.006), and a trend towards a lower number of apoptotic cells (p value 0.05).

Taking the drug-tolerant cells out of JQ1 into DMSO represents a change in culture condition. In this setting, there is no statistically significant difference in the percentage of apoptotic cells. However, there is a statistically significant difference in the percentage of necrotic cells in sensitive cells in DMSO and drug-tolerant cells (2.5% vs 8%, p value 0.01). However, the difference in the percentage of necrotic cells in drug-tolerant cells maintained in JQ1 (compared to those withdrawn from JQ1) is not statistically significant. Our transcriptomic data (Figure 3) suggests ongoing effects of BET bromodomain inhibition, even in drug-tolerant cells. Our hypothesis is that when drug-tolerant cells are withdrawn from JQ1, that associated changes in transcription may be toxic for a small number of cells (albeit not statistically significant), which is reflected in the results the Reviewer highlights.

We have amended the figure to depict the statistically significant differences in the percentage of apoptotic and necrotic cells.

The association between cell death and neuronal differentiation is intriguing and we thank the Reviewer for highlighting this. Programmed cell death has been shown to be an important aspect of early neuronal differentiation, while terminally differentiated neurons are less susceptible to cytotoxic stimuli(13). The continued ability of some drug-tolerant cells to undergo necrosis and apoptosis is thus consistent with our conclusion that the population of drug-tolerant cells have not undergone terminal neuronal differentiation. However, we refrained from speculating about this in the manuscript.

3. Supplementary Fig. 4: Figure legend should be revised (and reference in text) as the naming of F-G is different from the figure legend.

We have corrected the Figure legends.

4. Fig. 3C/D: The authors used a previously-described JQ1 transcriptional signature to classify drug-sensitive and tolerant cells. What were the differentially expressed genes between the JQ1-sensitive and tolerant cells using a naïve, unbiased approach, and how many of these genes overlapped with the previously-described JQ1 signature?

Thank you for suggesting this analysis, which further supports our conclusions that drug-tolerant cells reinstate expression of genes suppressed by BET-bromodomain inhibition in sensitive cells. We have added these analyses to the revised manuscript.

'We hypothesized that drug-tolerant cells evade BETi effects by reversing its transcriptional consequences. We performed genome-scale expression profiling of sensitive and drug-tolerant cells following treatment with DMSO or JQ1. We found 3,279 genes to be significantly upregulated in drug-tolerant cells cultured in JQ1 compared to drug-naïve cells (Supplemental Figure 8D, Supplemental Table 5A). These were significantly enriched for transcriptional targets of JQ1 in D458 cells (1667 genes, p value <0.0001). They were also enriched for genes that have previously been found to be suppressed by BETi both across cancers (JQ1 consensus signature(2, 11) Supplemental Table 5B, Figure 3C) and among medulloblastoma cell lines(2) (Figure 3D) (p value <0.0001 in both cases). However, removal of JQ1 increased expression of both of these genesets in drug-tolerant cells even further (Figures 3C and 3D, p<0.0001 for both genesets), suggesting residual JQ1 activity in these cells. These cells also exhibited increased expression of BET-bromodomain target pathways including MYC activation and E2F signaling(2, 11, 12) and of MYC itself (Supplemental Figure 9A and 9B). Taken together, these data suggest that the drug-tolerant D458 cells exhibit attenuated phenotypic and transcriptional responses to treatment with BETi.'

5. Fig. 3E/F: Quantitation should be provided for immunoblots a. It is not evident that BCL2L2, CCND2 and HLX are suppressed by JQ1 in drug-sensitive cells, as claimed in the text

We have now included quantification of the immunoblots in Supplemental Figure 9 and the Source Data File. Following quantification, suppression of CCND2 protein was statistically significant and there was a trend toward suppression for BCL2L1 and HLX, with p values of 0.05. We have amended text to reflect this:

'We next validated that rescue genes identified in our ORF screens, including cell-cycle regulators and bHLH transcription factors, were relevant in these models. Among the 18 rescue ORFs, five ORFs (BCL2L1, CCND2, HLX, NEUROD1 and NEUROG1) were either re-expressed in drug tolerance cells following suppression with BETi or exhibited increased expression with drug-tolerance. The cell-cycle regulator *CCND2* was suppressed by BETi (p value 0.04, Figure 3E and Supplemental Figure 9C), while the bHLH transcription factor *HLX* and the anti-apoptotic protein *BCL2L1* trended towards suppression with BETi (p value 0.05 and 0.059 respectively, Figure 3E and Supplemental Figure 9C-E). All three of these were re-expressed at both the mRNA and protein levels in drug-tolerant cells (Differential mRNA expression $q < 0.1$, Figure 3E and Supplemental Figures 9C-E and 11B (mRNA), p values 0.0002, 0.02 and 0.01 respectively), suggesting that these were 'mediator' genes whose expression had been reinstated. The other 15 rescue ORFs had not been suppressed by BETi, but expression of two of these also increased in drug-tolerant cells: the bHLH transcription factors NEUROD1 and NEUROG1 (Figure 3F, Supplemental Figures 9F and 9G, p values 0.0005 and 0.002 respectively for sensitive and drug-tolerant cells in DMSO).'

6. *Fig. 4A: Because BET inhibitors target K27ac, it would be good to include the K27ac profile by mass spectrometry, even if this histone mark is not differentially altered in drug-tolerant lines. Specific peptides with multiple PTM combinations are differentially found between sensitive and drug tolerant states and authors describe an overall trend towards activating PTMs specific to tolerant state and repressive PTMs in sensitive state.*

We agree with the Reviewer that a comprehensive examination of the changes in chromatin marks between drug-sensitive and drug-tolerant cells is important. The Global Chromatin Profiling performed included mass spectrometry profiling of 76 distinct histone marks, including K27ac. We now include the comparative marker selection analysis for all of these marks in Supplemental Table 8. We did not observe any statistically significant differences between the K27ac profiles of drug sensitive and drug-tolerant cells and now highlight this in the discussion.

We have added the following text to the results:

'We did not observe any statistically significant differences in the H3K27ac profiles of drug-sensitive and drug-tolerant cells.'

What is the overall fold change in sum abundance of marks such as H3K27me3 and H3K9me2/3?

The overall log fold change for in the drug-tolerant cells was -0.8 for H3K37me3, -0.7 for H3K9me2 and -0.9 for H3K9me3. These values (in addition to log fold changes for all histone marks) are now included in Supplemental Table 8.

7. *Page 14: "tolerant cells exhibited increased levels of the non-repressive mono and di-methyl forms of H3K27". The functions of H3K27me1/2 are not well characterized and it is not clear that these are non-repressive marks.*

We agree with the Reviewer that the significance of H3K27me1/2 has not been extensively characterized. We have therefore amended the sentence to remove reference to them being non-repressive marks. The sentence now reads:

'In contrast, drug sensitive cells had increased levels of the polycomb repressive marks H3K27me3 and the H3K9me3.'

8. *Fig. 4B/C: What were the changes in H3K4me3 and H3K27ac ChIP-binding at baseline (DMSO) in drug-resistant cells? All other graphs have this comparison*

We did not include drug-tolerant cells that had been withdrawn from JQ1 (and cultured in DMSO) in our ChIP-sequencing analysis because the 'baseline' state for these cells was maintenance of JQ1. Sensitive cells were cultured chronically in the presence of JQ1 until they exhibited features of tolerance such as ability to proliferate, reduced apoptosis and attenuated transcriptomic approaches.

9. *Fig. 4D: It is not clear based on ChIP tracks that there is increased H3K27ac at the promoter or enhancer of NEUROG1 in drug-tolerant cells*

We agree with the Reviewer that these changes are subtle at single gene level. We have therefore toned down our conclusions with respect to individual genes in the Results and removed the ChIP tracks from the figures.

'The changes in chromatin landscapes appear to facilitate the expression of rescue genes that we had observed in our drug-tolerant cell lines (and had scored in the D458 rescue screen, including cell-cycle regulators and bHLH/homeobox transcription factors). Rescue genes exhibited increased levels of total H3K4me3 in drug tolerant D458 medulloblastoma cells compared to sensitive cells in either DMSO (p value 0.03) or JQ1 (0.02, Figure 4B). Treatment of D458 cells with JQ1 was associated with increased levels of total H3K27ac at rescue genes (p value 0.0002, Figure 4C), which persisted in drug tolerant cells, which maintained elevated levels of total H3K27ac at rescue genes relative to untreated D458 cells (p value 0.01, Figure 4C). There was a trend for D458 drug-tolerant cells to exhibit increased levels of H3K27ac binding at rescue genes compared to sensitive cells treated with JQ1, however this did not reach statistical significance (p value 0.05, Figure 4C). We did not observe similar changes in relation to genes that did not score as rescue genes in our ORF screens (Supplemental Figure 11E). In aggregate, these data suggest that drug tolerance involves activation of promoter- and enhancer-associated marks preferentially for rescue genes.'

10. *Supplementary Fig. 6A: Quantitation of immunoblot is necessary*
a. *Why is TUJ1 band of a different size in NEUROG1-expressing cells?*

We now include densitometry measurements and relevant statistics in Supplementary Figure 9H.

We do not know the source of the subtle difference in size of TUJ1 in neural stem cells expressing NEUROG1 (and NEUROD1). We include as Reviewer Figure 1 images of two other independent replicates of this experiment. These show TUJ1 as a doublet, suggesting TUJ1 undergoes post-translational modifications. In these replicate experiments, the higher band is slightly stronger in the NEUROD1 and NEUROG1 expressing cells, suggesting variation in the degree of post-translational modification. While interesting, we have not yet followed up on this phenomenon as it is outside the scope of this manuscript.

11. *Fig. 5A: Format of immunoblot is misleading: the DMSO control should be on the left and JQ1-treated to the right in drug-tolerant cells, as was done throughout the paper*

We have replaced this immunoblot with that of a replicate experiment to ensure it is consistent with the other immunoblots throughout the manuscript.

12. *Page 16: "The drug-tolerant cells also had increased expression of neuronal markers NEUROG1, NEUROD1, TUJ1 and NF68"*
Which figure shows increased expression of NEUROG1, NEUROD1 and TUJ1 in drug-tolerant cells?

NEUROG1 and NEUROD1 were included in Figure 3F and we have corrected the citation. We sincerely apologize as we realize that the TUJ1 western was omitted during revisions for the prior submission. We now

show TUJ1 protein expression in Figure 4D. Densitometry measurements across replicate experiments are included in Supplemental Figures 9 and 10.

13. ChIP normalization by spike in would benefit the quantitative description of changes in H3K27ac over targets described to be implicated in JQ1 resistance.

We agree that ChIP normalization is important. While we did not utilize a ‘spike in’ approach, we included a total H3 input control for each sample, which was used to normalize each of the different histone ChIPs for that sample. We also applied standard MACs peak calling approaches for data normalization. In addition, when calculating ChIP-binding scores at a gene level, we calculated gene level Z-scores within samples to allow comparison across other samples.

14. Does relaxation of stringency thresholds (p value range) of gene expression data and Rescue genes from ORF screen increase the number of overlapping hits? Where is CCND2 ranked in overall datasets for D238 line? Do you find more members of pathways of interest (BCL2 family, developmental genes) when thresholds are relaxed?

We have repeated this analysis by using a more relaxed threshold for Rescue genes from the ORF screens (log fold change >1.1 and q value <0.25).

For the D458 cell line, this increased the number of ORF hits to 35, seven of which were also identified as genes that are suppressed by BETi (previously three genes). This expanded list did indeed include more members of the pathways of interest, including BCL2 family members BCL2, BCL2L1, and BCL2L2; cell-cycle members CCND1, CCND2, CCND3 and CCNE2; and bHLH or homeobox transcription factors including HESX1, HLX, HOXC11, LHX6, MSX2, NEUROG1, NEUROG3, SIX2, SPN and TLX3.

The relaxed threshold increased the number of D283 associated ORF hits to 40, 12 of which were also identified as genes that are suppressed by BETi (previously five genes). As with D458, the number of genes across the BCL2 family members and bHLH or homeobox transcription factors that were identified as hits was increased. Furthermore, CCND2 was identified as a hit (in the IBET treated replicates) using this relaxed threshold. In our previous analysis, while CCND2 met statistical significance in the IBET replicates, it did not meet the log fold change threshold of 1.5 (average log fold threshold of 1.4).

The genes identified as hits with the relaxed ORF threshold are now included in Supplemental Table 4.

Reviewer #3 (Remarks to the Author):

This manuscript focuses on the mechanisms of resistance to BET bromodomain inhibitors (BETi) in medulloblastoma (MB). Using a combination of RNAseq to identify genes that are downregulated by BETi, a CRISPR screen to discover genes that are essential for cell survival, and an ORF overexpression screen to find genes that can rescue cells from the effects of BETi, they come up with a list of genes that are likely to be key mediators of BET inhibition. These genes include regulators of the cell cycle such as D-cyclins, regulators of apoptosis such as BCL2 family members, and bHLH transcription factors (including members of the NeuroD and NeuroG families) that regulate cell fate and differentiation. They show that several of these genes are upregulated in cells that have become resistant to BET inhibitors, and that this upregulation is a consequence of epigenetic reprogramming. Finally, they demonstrate that counteracting the upregulation of cyclins by treating cells with CDK4/6 inhibitors can overcome resistance and synergize with BET inhibitors to antagonize tumor growth. Based on these studies, they propose that combination therapy with BET inhibitors and CDK4/6 inhibitors will be effective for MYC-driven MB.

This paper is a technical tour-de-force, combining an array of cutting edge functional genomic approaches to elucidate the mechanisms of BETi activity. Overall, the experiments are well designed and controlled, and the

conclusions are well supported by the data. My main concern is that many of the experiments are not described with sufficient clarity and in sufficient detail to be comprehensible to the average reader. There are also some instances where the figures are not clearly presented, or where the data are not presented in figures at all. Most of these issues should be addressable with revisions to the text and figures, and with a few fairly straightforward experiments.

We thank the Reviewer for their positive comments about our manuscript. We agree that the multiple genomic approaches and assays presented do make it difficult for the reader to follow. We have attempted to expand our descriptions of the experiments as outlined below.

1. Most of these studies were done on cell lines that have been in culture for several decades. It is understandable that some of the approaches – eg. CRISPR screening, ORF screening – would be challenging to carry out using PDXs or mouse models, but the authors should at least discuss the limitations of the cells they are using, and if possible, perform some of the validation studies on patient-derived xenografts that more closely resemble the human disease.'

We agree that validating results across cell lines is important. Our study included five distinct MYC-driven medulloblastoma cell lines that have been shown to recapitulate the human disease(1). This included the recently generated MB002 cell line, which was established in 2012 by Dr. Cho at Stanford. This line is not passaged in serum and has been shown to recapitulate Group 3 medulloblastomas(2).

Our revised manuscript includes validation in a further three lines, including in vitro combination testing of a model generated from our collaborators in Heidelberg(3), and in vivo testing of two patient derived xenograft models generated by our collaborator Dr. Olson at the Fred Hutchinson Cancer Research Center(4).

2. The authors state (p. 6) that they performed expression profiling on 5 MYC-driven MB lines, but in Supplemental Table 1 they only present data on 4 lines.

We apologize for this confusion. We were referring to the gene expression profiling that we had performed across five MYC-driven lines and published in our CCR paper(2). However, we have also performed additional profiling of four lines which form the basis for the gene-expression analysis presented in this manuscript. We have amended text to refer to expression profiling to these four MYC-driven lines.

In addition, few details are given about this experiment: what concentration of BETi was used, and how long were cells treated with BETi before gene expression was analyzed? Some of this information is in the Methods section, but it really should be spelled out in the text and/or legends so readers can follow what was done.

We have now included the details of the concentration of JQ1 used, and the time point at which the mRNA was extracted in the Figure Legend as shown below:

'A. Intersection of genes suppressed 24 hours following treatment with 1 μ M JQ1, relative to DMSO controls (blue), with those found to be cell essential (green) in D458 (top) and D283 (bottom) medulloblastoma cell lines by CRISPR-Cas9 screens. P-values indicate significance of overlap.'

Also, it is not clear how effective JQ1 was at inhibiting growth/survival in the cells used here. Even if similar data were shown in a previous paper, it would be helpful to show growth or survival curves here so that readers can appreciate the effects for which a mechanistic explanation is being sought.

We agree that including more efficacy data for each of the lines would help the readership. We have now included the JQ1 response data in Supplemental Figures 13 and 14. In addition, we have included JQ1 only controls in each of the in vitro and in vivo experiments evaluating combination treatment.

Finally, the gene expression data are discussed very briefly, and not presented anywhere except in Supplementary Table 1. It would be helpful to see some kind of graphical representation of these data, e.g. a heat map, a cytoscape plot of the pathways that were altered, etc.

We now include heatmaps showing the 50 most differentially expressed genes in each of the four cell lines in Supplemental Figures 1 and 2.

3. The CRISPR screen is also poorly described in the text. How were cells cultured, and for how long? What assay was used to determine dependency, and what were the criteria for a gene to be considered a dependency? Again, this information might be in the methods, but it is essential to provide at least some detail in the text and in the figure/table legends so that readers can follow the experiment.

In response to the Reviewer's suggestion, we have added the following text to the Results section:

'We therefore determined which of the suppressed genes are cell-essential. We applied a pooled CRISPR/Cas9 screen targeting 18,454 genes (Supplemental Figure 1A) to each of the D458 and D283 cell lines. Cells were first infected with Cas9 and then infected with the pool guide RNA (sgRNA) lentiviral libraries with a low multiplicity of infection (30-50% infection efficiency). After selection, cells were cultured for 21 days before DNA was harvested. DNA barcodes labeling each sgRNA were PCR amplified and next generation sequencing was performed to determine relative abundance of each sgRNA. Essential genes were those that were depleted at the end of the assay relative to the early time point.'

And Figure Legend 2A 'Cell-essential genes were identified as genes that were depleted following infection with the pooled lentiviral guide library. Abundance of each sgRNA guide (labeled with a unique DNA barcode) was measured at the initiation and completion of each assay (via PCR amplification and next-generation sequencing). Genes with a dependency probability of >0.35 with a FDR <0.2 were deemed to represent essential genes within each line. P-values indicate significance of overlap between genes that are suppressed by BETi and cell-essential genes.'

The Methods section also includes an expanded description of the methodology, including details of data analysis including how dependency probabilities were calculated.

Finally, the authors state (p. 7) that the dependencies were enriched for members of 39 pathways, but no information is provided about how these pathways were identified.

We apologize for this omission and have now included the following text in the Figure Legend to describe Supplemental Table 3:

'Pathways enriched in genetic dependencies that are exploited by BETi. Among genes that are cell-essential, we applied Gene Set Enrichment Analysis and the C2CP pathways database to identify pathways that were enriched with genes found to be suppressed by BETi.'

The pathways are shown in Supp. Table 3, but the genes in these pathways are not listed, which makes it difficult to evaluate their significance. This information should be added to the table.

We have now added these genes to Supp Table 3.

Some graphical representation of the CRISPR screen results would also be helpful.

Thank you for this suggestion. We now include plots to demonstrate the thresholds that were used to define 'essential genes' (Supplemental Figure 4), in addition to figures that depict the 50 most essential cell lines within each cell line (Supplemental Figure 5).

4. In describing the ORF screen, the authors state that they “applied a lentivirally delivered ORF library...” in cell lines treated with BETi or vehicle, but they do not describe anything else about the assay or the criteria for determining whether a gene rescued cells from BETi. They state that “Log-fold-changes for each gene were highly correlated...”, but it is not clear what was being measured (gene expression?) and what was being compared to what to determine fold change. These are not simple or standard assays, and the reader needs to be brought along for the ride, rather than being asked to simply accept the conclusions.

We have included the following text to the Results section and the Figure Legend describing the ORF screen:

Results section: ‘We therefore attempted to narrow the set of essential genes and pathways that BETi exploits to those that are required for BETi phenotypic effects, by determining which genes were sufficient to rescue cells from BETi. We applied a lentivirally delivered ORF library encompassing 12,579 genes, in both the D458 and D283 cell lines, each treated with either of two structurally distinct BET-bromodomain inhibitors (JQ1 and IBET151) or vehicle control (See Methods and Supplementary Figure 1C). Abundance of each ORF was measured at the initiation and completion of each assay to determine log-fold changes following treatment with BET-bromodomain inhibitors. We defined “rescue ORFs” as those conferring >1.5 log-fold enrichment with a q value of <0.25.’

and

Figure legend 2B: ‘Rescue genes identified in D458 (left) and D283 (right) cell lines following treatment with either JQ1 or IBET151. Abundance of each ORF (labeled with a unique DNA barcode) was measured at the initiation and completion of each assay (via PCR amplification and next-generation sequencing) to determine log-fold changes following treatment with BET-bromodomain inhibitors. We defined “rescue ORFs” as those conferring >1.5 log-fold enrichment with a q value of <0.25. Asterisks indicate genes that scored as statistically significant rescue genes in both cell lines, but only met fold-change thresholds in the cell line shown. P-value indicates significance of overlap.’

5. The downstream analysis of the ORF screen results is also not very clearly explained. “Proteins related to cell fate commitment, transcription and developmental processes were significantly enriched in the rescue gene network”: what is the rescue gene network and how was it derived? “This pathway enrichment was reflected by a high connectivity for the ORF network as a whole, with 31 nodes and 42 edges” What does high connectivity mean? What are nodes and edges?

We have expanded our description of this analysis in both the Results section and the Figure Legend as below:

Results: ‘Proteins related to cell fate commitment, transcription, and developmental processes were significantly enriched in the rescue gene network ($q < 0.0001$), as were MYC-type basic helix-loop-helix (bHLH) ($q < 0.001$), cyclin ($q < 0.01$), and myogenic basic muscle-specific protein domains ($q < 0.001$) (Supplemental Figure 1F). We performed protein network analysis using String (see Methods) on the 31 ORFs that had been identified as rescue genes. When considering these ORFs to be nodes and edges to represent protein-protein interactions between those nodes, we determined that this network had 42 edges. This was much higher than the expected number of edges in a network with randomly chosen nodes (8 edges, $p < 0.0001$, Figure 2C), indicating high connectivity within this network.

Figure Legend 2C: STRING(14) protein network analysis was applied to identify direct and functional protein networks that exist between the entire set of candidate rescue ORFs identified across both cell lines. Edges indicate protein-protein interactions between ORF rescue genes (represented as nodes) that scored in either D458 or D283 following treatment with JQ1 or IBET151. This network had significantly higher connectivity than expected, as shown by the p value, which reflects the likelihood of attaining this number of edges in a randomly selected protein-protein interaction network with the same number of nodes.

6. *The low-throughput rescue assays should also be clarified. What assay was used to determine rescue, and what were the criteria for a gene to be considered to have rescued cells from JQ1? Also, in describing the results of these assays, the authors mention CCND2 and CCND3 rescuing D458 cells, but they do not mention BCL2L1, which appears to be significant based on Figure 2E.*

We have now added further experimental details to the Results section as requested by the Reviewer. We also thank the Reviewer for catching our failure to mention BCL2L1 which we have corrected:

‘We validated these genes in low-throughput assays (Figures 2E and F, Supplemental Figures 6A-C and Supplemental Figure 7). We transduced medulloblastoma cells with lentivirus containing pLEX-307 lentiviral vectors, to overexpress eGFP, CCND2, CCND3, BCL2L1, MYOD1, MYOG, NEUROD1, NEUROG1 and NEUROG3, in independent infections. Cells were selected for 48 hours and then treated with 1 μ M of JQ1 (or DMSO control) for 11 (D283) or 14 (D458) days. Over-expression of CCND2, CCND3 and BCL2L1 rescued D458 cells from the effects of JQ1 (p values 0.002, 0.002 and 0.01) and CCND3 and NEUROG1 rescued D283 cells (p value =0.002 and 0.01). There was a trend for over-expression of CCND2 and BCL2L1 in D283 to confer selective advantage in JQ1, but these did not reach statistical significance (p=0.08 and 0.06 respectively). We also validated additional bHLH transcription factors as rescue genes: MYOD1 in D458 (p=0.04) and NEUROD1 (p=0.027), NEUROG1 (0.02) and NEUROG3 (p=0.02) in D283. Overexpression of these ORFs did not confer growth advantages in any of the cell lines when passaged in DMSO (and indeed, were associated with attenuated growth in some instances; Supplemental Figure 7). Expression of BCL2L1 and NEUROG3 attenuated JQ1-induced apoptosis relative to eGFP controls in both D458 and D283 (p values D458 BCL2L1 0.085 and NEUROG3 0.012; D283 BCL2L1 <0.0001 and NEUROG3 0.0017, Figure 2F), as did CCND2 and NEUROD1 in D283 cells (p values 0.0028 and <0.0001 respectively).

We also validated these ORFs as rescue genes in other patient-derived MYC-driven medulloblastoma cell lines (D341 and two that are passaged in serum-free conditions: CHLA01 and the recently generated cell line MB002). In each line, at days seven post treatment, we found responses to JQ1 were attenuated by overexpression of a cell cycle regulator, BCL2 family member, and at least one bHLH/homeobox transcription factor (Supplemental Figure 6C and Supplemental Figure 7). Thus, while we observed cell-specific differences in the magnitude of resistance for individual ORFs, we observed consistency in drug resistance at the pathway level (i.e. cell cycle, apoptosis avoidance, and bHLH/homeobox transcription factors).’

And Figure Legend:

‘Low throughput rescue assays in D458 and D283 cells expressing eGFP, CCND2, CCND3, BCL2L1, NEUROD1, NEUROG3, MYOD1 or MYOG that were treated with JQ1 1 μ M or DMSO control. Cumulative doublings of cells passaged in JQ1 for 11-14 days, relative to DMSO controls. Error bars depict mean \pm SEM. Asterisks denote significant differences from eGFP controls (* p<0.05, ** p<0.01, *** p<0.001) as determined by Two-tailed unpaired t-tests. Source data are provided as a Source Data File.’

7. *On the top of p. 11, they refer to Supplemental Figure 2E, but there is no panel E in Supp. Fig. 2.*

We have corrected this citation to refer to Supplemental Figures 6A-C and Supplemental Figure 7

8. *The authors state (p. 11) that they sought to determine if rescue genes were differentially expressed in “naturally arising” models of BETi resistance. The term naturally arising is somewhat misleading: treating a long-term passaged cell line with an inhibitor for a month is not at all natural. The authors should use another phrase to describe these experiments.*

We agree with the Reviewer and have amended the text as follows:

'We next sought to determine if the rescue genes identified in our ORF screens were differentially expressed in medulloblastoma cells that acquire tolerance to BETi. We therefore passaged D458 cells and the related D425 line¹⁶ in JQ1 until they exhibited growth in the presence of increasing concentrations of JQ1 and IBET151 (cells were cultured in the presence of BETi for greater than 30 days in each case; Supplemental Figure 8A). We were unable to isolate drug-tolerant cells from the other medulloblastoma cell lines.'

9. In discussing Figure 3 the authors state that they probed expression profiles of BETi sensitive and tolerant cells for previously described signatures of JQ1 transcriptional response. The genes they used for these signatures should be included in a Supplemental Table. They also state that expression of JQ1 signature genes increased following removal of JQ1, but they don't cite a figure or table in which these data are presented.

Thank you for this suggestion. We now include the genes associated with the JQ1 consensus signature in Supplemental Table 5. We also cite Figure 3C which demonstrates that the expression of JQ1 signature genes increase following the removal of JQ1.

'We hypothesized that drug-tolerant cells evade BETi effects by reversing its transcriptional consequences. We performed genome-scale expression profiling of sensitive and drug-tolerant cells following treatment with DMSO or JQ1. We found 3,279 genes to be significantly upregulated in drug-tolerant cells cultured in JQ1 compared to drug-naïve cells (Supplemental Figure 8D, Supplemental Table 5A). These were significantly enriched for transcriptional targets of JQ1 in D458 cells (1667 genes, p value <0.0001). They were also enriched for genes that have previously been found to be suppressed by BETi both across cancers (JQ1 consensus signature^{5,6} Supplemental Table 5B, Figure 3C) and among medulloblastoma cell lines⁶ (Figure 3D) (p value <0.0001 in both cases). However, removal of JQ1 increased expression of both of these genesets in drug-tolerant cells even further (Figures 3C and 3D, p<0.0001 for both genesets), suggesting residual JQ1 activity in these cells. These cells also exhibited increased expression of BET-bromodomain target pathways including MYC activation and E2F signaling⁴⁻⁶ and of MYC itself (Supplemental Figure 9A and 9B). Taken together, these data suggest that the drug-tolerant D458 cells exhibit attenuated phenotypic and transcriptional responses to treatment with BETi.'

10. On p. 13, they state that expression of NEUROD1 and NEUROG1 "also increased with treatment of BETi". This seems to contradict previous statements that these genes decreased following BETi treatment. Do the authors mean that these genes decrease in cells that are resistant to BETi?

We apologize that this is confusing. We observe variability in the specific bHLH transcription factors that are suppressed in the cell lines. For the D458 cell line, HLX met our criteria of being a JQ1 transcriptional target (was suppressed in gene-expression profiling) that was re-expressed in drug-tolerant lines. In this line, NEUROD1 and NEUROG1 were not suppressed with treatment with BETi, but the genes scored as rescue genes in the ORF screen, and D458 drug-tolerant cells exhibit increased expression of these transcription factors.

In the D283 cell line, NEUROD1 and NEUROG1 also scored as rescue genes, but in contrast to D458 these genes were both also transcriptional targets of BETi. We have been unable to generate drug-tolerant D283 lines.

We describe these results in the paragraphs below:

‘Integrating all three datasets—gene expression, CRISPR/Cas9 screen, and ORF rescue—cell-cycle genes (*CCND2* and *CCND3*) scored in D458 and the anti-apoptosis gene *BCL2L1* and bHLH transcription factor-encoding gene *NEUROG1* also scored in D283 (Figure 2D). In addition, the cell-cycle gene *CCND2* scored as a dependency gene that is suppressed by JQ1 in D283 but only met the q-value (not log fold-change) threshold for a rescue gene.’

and

‘We next validated that rescue genes identified in our ORF screens, including cell-cycle regulators and bHLH transcription factors, were relevant in these models. Among the 18 rescue ORFs, five ORFs (*BCL2L1*, *CCND2*, *HLX*, *NEUROD1* and *NEUROG1*) were either re-expressed in drug tolerant cells following suppression with BETi or exhibited increased expression with drug-tolerance. The cell-cycle regulator *CCND2* was suppressed by BETi (p value 0.04, Figure 3E and Supplemental Figure 9C), while the bHLH transcription factor *HLX* and the anti-apoptotic protein *BCL2L1* trended towards suppression with BETi (p value 0.05 and 0.059 respectively, Figure 3E and Supplemental Figure 9C-E). All three of these were re-expressed at both the mRNA and protein levels in drug-tolerant cells (Differential mRNA expression $q < 0.1$, Figure 3E and Supplemental Figures 9C-E and 11B (mRNA), p values 0.0002, 0.02 and 0.01 respectively), suggesting that these were ‘mediator’ genes whose expression had been reinstated. The other 15 rescue ORFs had not been suppressed by BETi, but expression of two of these also increased in drug-tolerant cells: the bHLH transcription factors *NEUROD1* and *NEUROG1* (Figure 3F, Supplemental Figures 9F and 9G, p values 0.0005 and 0.002 respectively for sensitive and drug-tolerant cells in DMSO).’

11. On p. 16, the authors state that “The drug-tolerant cells also had increased expression of neuronal markers NEUROG1, NEUROD1, TUJ1 and NF68, and suppression of the stem marker MSI1.” The only Figure cited in this section is Figure 5, and while this figure shows data for NF68 and MSI1, there are no data for NEUROG1, NEUROD1 or TUJ1. Are those genes presented elsewhere?

NEUROG1 and NEUROD1 were included in Figure 3F and we have corrected the citation. We sincerely apologize as we realize that the TUJ1 western was omitted during revisions for the prior submission. We now show TUJ1 protein expression in Figure 4D. Densitometry measurements across replicate experiments are included in Supplemental Figures 9 and 10.

12. The authors hypothesize that resistance to BETi is associated with acquisition of a cellular state that is primed for differentiation, and later show that resistance to BETi is predetermined. These concepts – that the resistant state is acquired and that it is pre-existent – are quite different. The authors should discuss this more extensively and try to reconcile them into a single coherent hypothesis. Moreover, their description of the more differentiated resistant cells sounds a lot like what some investigators refer to as “transit amplifying cells”. The authors should explore this concept in their discussion.

We agree with the Reviewer that the terms ‘acquired’ and ‘pre-existing’ are confusing. In using the word ‘acquired’, we were referring to changes that were differentially present in the drug-tolerant cells relative to sensitive cells. However, these can also be pre-existing alterations in that if ‘sensitive cells’ that do not harbor high-levels of expression of, for example, rescue genes, preferentially die in response to BETi, the surviving drug-tolerant cells will have higher levels of expression of the rescue gene. We have added the following sentence to introduce the section describing the barcoding and flow cytometry experiments:

‘The finding that drug-tolerant D458 cells differed from treatment-naïve D458 cells in their cell state, expression profiles, and response to BETi could result from either individual cells acquiring these phenotypes or from selection of a subpopulation of cells that exhibited these phenotypes prior to treatment.’

We also thank the Reviewer for raising the concept that the features of the drug-tolerant cells are similar to 'transit amplifying cells', or 'activated quiescent neural progenitors' as they have been referred to in the neural context(16). In response to both this and the prior comment, we have now added the following paragraph to the Discussion:

'Brain tumors are comprised of a hierarchy of subpopulations of cells, ranging from those that resemble stem cells with self-renewal capacity to those in which lineage commitment has been triggered through the expression of transcription factors(6-8). Our analysis of stem and lineage markers support similar heterogeneity within (and between) medulloblastomas, a finding that has also been reported in single cell RNA-sequencing profiles of human medulloblastomas(5). We identified cells that express both genes associated with neuronal differentiation and stemness as being associated with drug-tolerance, and found similar cells to be present in both pretreatment medulloblastoma cell lines and primary tumors, raising the possibility that selection of these intrinsically drug-tolerant cells contribute to BETi tolerance. Similar populations of cells have been identified in other normal developmental hierarchies, and in cancer, and have been termed transit amplifying cells(9, 10) or in the neural context, 'activated quiescent neural progenitor cells'. These cells have been shown to harbor a phenotype that is more differentiated than stem cells, and to harbor increased proliferate potential compared to quiescent stem cells or terminally differentiated neurons.'

13. After an extraordinary amount of functional genomics and epigenomics, the authors identify bHLH transcription factors, BCL2 family members and D-Cyclins as key mediators of BETi and of resistance to these drugs. Among these, they focus on D-Cyclins, and test the efficacy of the CDK4/6 inhibitor LEE011 (ribociclib) as a strategy for overcoming BETi resistance. Although ribociclib has not been extensively studied in MB, the related agent palbociclib has. The authors should cite the previous papers on palbociclib (Cook Sangar et al., Clin Cancer Res 2017; Hanaford et al., Clin Cancer Res 2016; Whiteway et al., J. Neurooncol 2013) and discuss their results in the context of those papers.

We agree with the Reviewer that the previous studies of cell-cycle inhibitors are highly relevant as they have paved the path for early phase pediatric clinical trials for CDK4/6 inhibitors. We include the following paragraph in our Discussion, citing the papers referenced by the Reviewer:

'Furthermore, the observation that BETi-tolerance is pre-determined in medulloblastoma cells provides rationale for initiating combination therapies to ablate these subpopulations of cells in BETi treatment naïve cells. Indeed, we observe BETi combination treatment with the CDK4/6 inhibitor LEE011 to attenuate the acquisition of drug-tolerance. Cell-cycle inhibition with CDK4/6 inhibitors including Palbociclib have been shown to have preclinical promise in models of MYC-amplified medulloblastoma(17-19), paving the way for early phase clinical trials for children with medulloblastoma. The combination of CDK4/6 and BET-bromodomain inhibition has also been reported to be synergistic in other contexts(20), while in medulloblastoma, BETi has also been shown to synergize with inhibition of CDK2(21).'

14. It is unfortunate that the authors did not test the functional importance of the other two classes of genes they identified: bHLH transcription factors and BCL2 family members. Although there are not specific drugs that can be used to inhibit bHLH transcription factors, knocking them down and testing the effects on resistance to BETi would enhance the novelty and significance of these studies. Similar knockdowns, or pharmacological inhibitors such as Navitoclax or Venetoclax or APG-1252, could also be informative regarding the importance of BCL2 family members.

We agree with the Reviewer that our finding that three pathways mediate both response and resistance to BETi is of interest. We now include experiments in which we have leveraged short hairpin RNA technology to suppress expression of these pathways (BCL2L1, CCND2 and NEUROD1) in both sensitive and drug tolerant cells. These data are included in Figures 3, Supplemental Figure 11 and Results section as below

'To further evaluate the functional significance of the anti-apoptosis, cell cycle and bHLH transcription factors in drug-tolerance, we suppressed BCL2L1, CCND2 and NEUROD1 using short hairpin RNAs,

confirming each of these genes to be essential in both D458 drug-naïve and -tolerant cells (p values <0.0001 in all cases, Figures 3G, 3H and Supplemental Figures 11C-11D). We concluded that rescue genes from cell-cycle, bHLH transcription factor, and anti-apoptotic pathways are cell-essential in D458 cells and are re-expressed in cells that acquire drug-tolerance, remaining genetic dependencies in those cells.'

15. *The authors state that they observed synergy between JQ1 and LEE011, but based on Supp. Fig. 6 they only tested two concentrations of LEE011 and one concentration of JQ1. It is unclear how they selected these particular concentrations, and what would have happened at other concentrations. The authors should use a more rigorous approach to test synergy, with multiple concentrations of each drug, such as the approach described in Chou et al., Cancer Res. 2010.*

We thank the Reviewer for this suggestion. We now include expanded testing of the *in vitro* testing of the combination of JQ1 and LEE011, across multiple concentrations of each drug, with formal testing for synergy. This data is included in the Results section and Supplemental Figures 12 and 13.

'We first evaluated the acute efficacy of the combination of JQ1 and LEE011 in cell lines that exhibit sensitivity to JQ1. We found this combination to meet BLISS criteria for synergy²⁷ in the D458, MB002 and D341 cell lines (Supplemental Figure 12B-F). We further validated synergy (using the LOEWES model for synergy) between LEE011 and JQ1 across a wider range of concentrations in D458 (p value <0.0001), MB002 (p value <0.0001) and D283 (p value <0.0001), along with an additional MYC-driven line HD_MB003 (p value <0.0001, Supplemental Figures 13 and 14). With prolonged treatment of D458 cells, we also found the addition of LEE011 to JQ1 delayed the acquisition of drug tolerance relative to treatment with JQ1 alone (Figure 5A).'

16. *Although the combination of JQ1 and LEE011 is more effective than either drug alone, Figure 5G shows that mice treated with the combination still succumb to their disease. The authors should discuss why this might be happening. When tumors are harvested from dually-treated animals, are they now resistant to both drugs? If so, what strategies could be used to overcome this resistance? Would inhibition of BCL2L1 or one of the bHLH proteins help abolish resistance?*

We agree with the Reviewer that understanding the resistance mechanisms to the combination is important, and we have plans for future studies to address this. Our shRNA experiments indicate that the drug tolerant lines require expression of all three pathways (cell-cycle regulators, BCL2 members and the bHLH transcription factor NEUROD1). In addition, expression of each gene results in partial rescue of the BETi phenotype. Thus, future strategies to target all of these pathways may help further abolish resistance. We have added this discussion to the manuscript:

'Furthermore, the observation that BETi-tolerance is pre-determined in medulloblastoma cells provides rationale for initiating combination therapies to ablate these subpopulations of cells in BETi treatment naïve cells. Indeed, we observe BETi combination treatment with the CDK4/6 inhibitor LEE011 to attenuate the acquisition of drug-tolerance. Cell-cycle inhibition with CDK4/6 inhibitors including Palbociclib have been shown to have preclinical promise in models of MYC-amplified medulloblastoma(17-19), paving the way for early phase clinical trials for children with medulloblastoma. The combination of CDK4/6 and BET-bromodomain inhibition has also been reported to be synergistic in other contexts(20), while in medulloblastoma, BETi has also been shown to synergize with inhibition of CDK2(21). We have also identified bHLH/homeobox transcription factors and BCL2 family members as mediating response and resistance to BETi. Future work will examine whether also inhibiting these pathways (in particular BCL2 family members) may represent a therapeutic strategy to attenuate resistance to the combination of cell-cycle inhibition and BETi.'

References

1. S. D. Weeraratne, V. Amani, N. Teider, J. Pierre-Francois, D. Winter, M. J. Kye, S. Sengupta, T. Archer, M. Remke, A. H. C. Bai, P. Warren, S. M. Pfister, J. A. J. Steen, S. L. Pomeroy, Y.-J. Cho, Pleiotropic effects of miR-183~96~182 converge to regulate cell survival, proliferation and migration in medulloblastoma, *Acta Neuropathol* **123**, 539–552 (2012).
2. P. Bandopadhyay, G. Bergthold, B. Nguyen, S. Schubert, S. Gholamin, Y. Tang, S. Bolin, S. E. Schumacher, R. Zeid, S. Masoud, F. Yu, N. Vue, W. J. Gibson, B. R. Paoletta, S. S. Mitra, S. H. Cheshier, J. Qi, K.-W. Liu, R. Wechsler-Reya, W. A. Weiss, F. J. Swartling, M. W. Kieran, J. E. Bradner, R. Beroukhim, Y.-J. Cho, BET bromodomain inhibition of MYC-amplified medulloblastoma, *Clin. Cancer Res.* **20**, 912–925 (2014).
3. T. Milde, M. Lodrini, L. Savelyeva, A. Korshunov, M. Kool, L. M. Brueckner, A. S. L. M. Antunes, I. Oehme, A. Pekrun, S. M. Pfister, A. E. Kulozik, O. Witt, H. E. Deubzer, HD-MB03 is a novel Group 3 medulloblastoma model demonstrating sensitivity to histone deacetylase inhibitor treatment, *J. Neurooncol.* **110**, 335–348 (2012).
4. S. Brabetz, S. E. S. Leary, S. N. Gröbner, M. W. Nakamoto, H. Seker-Cin, E. J. Girard, B. Cole, A. D. Strand, K. L. Bloom, V. Hovestadt, N. L. Mack, F. Pakiam, B. Schwalm, A. Korshunov, G. P. Balasubramanian, P. A. Northcott, K. D. Pedro, J. Dey, S. Hansen, S. Ditzler, P. Lichter, L. Chavez, D. T. W. Jones, J. Koster, S. M. Pfister, M. Kool, J. M. Olson, A biobank of patient-derived pediatric brain tumor models, *Nat. Med.* **24**, 1752–1761 (2018).
5. M. C. Vladoiu, I. El-Hamamy, L. K. Donovan, H. Farooq, B. L. Holgado, V. Ramaswamy, S. C. Mack, J. J. Lee, S. Kumar, D. Przelicki, A. MichaelRaj, K. Juraschka, P. Skowron, B. Luu, H. Suzuki, S. A. Morrissy, F. M. Cavalli, L. Garzia, C. Daniels, X. Wu, M. A. Qazi, S. K. Singh, J. A. Chan, M. A. Marra, D. Malkin, P. Dirks, T. Pugh, F. Notta, C. L. Kleinman, A. Joyner, N. Jabado, L. Stein, M. D. Taylor, Childhood cerebellar tumors mirror conserved fetal transcriptional programs, *bioRxiv*, 350280 (2018).
6. R. J. Vanner, M. Remke, M. Gallo, H. J. Selvadurai, F. Coutinho, L. Lee, M. Kushida, R. Head, S. Morrissy, X. Zhu, T. Aviv, V. Voisin, I. D. Clarke, Y. Li, A. J. Mungall, R. A. Moore, Y. Ma, S. J. M. Jones, M. A. Marra, D. Malkin, P. A. Northcott, M. Kool, S. M. Pfister, G. Bader, K. Hochedlinger, A. Korshunov, M. D. Taylor, P. B. Dirks, Quiescent sox2(+) cells drive hierarchical growth and relapse in sonic hedgehog subgroup medulloblastoma, *Cancer Cell* **26**, 33–47 (2014).
7. I. Tirosh, A. S. Venteicher, C. Hebert, L. E. Escalante, A. P. Patel, K. Yizhak, J. M. Fisher, C. Rodman, C. Mount, M. G. Filbin, C. Neftel, N. Desai, J. Nyman, B. Izar, C. C. Luo, J. M. Francis, A. A. Patel, M. L. Onozato, N. Riggi, K. J. Livak, D. Gennert, R. Satija, B. V. Nahed, W. T. Curry, R. L. Martuza, R. Mylvaganam, A. J. Iafrate, M. P. Frosch, T. R. Golub, M. N. Rivera, G. Getz, O. Rozenblatt-Rosen, D. P. Cahill, M. Monje, B. E. Bernstein, D. N. Louis, A. Regev, M. L. Suvà, Single-cell RNA-seq supports a developmental hierarchy in human oligodendroglioma, *Nature* (2016), doi:10.1038/nature20123.
8. M. G. Filbin, I. Tirosh, V. Hovestadt, M. L. Shaw, L. E. Escalante, N. D. Mathewson, C. Neftel, N. Frank, K. Pelton, C. M. Hebert, C. Haberler, K. Yizhak, J. Gojo, K. Egervari, C. Mount, P. van Galen, D. M. Bonal, Q.-D. Nguyen, A. Beck, C. Sinai, T. Czech, C. Dorfer, L. Goumnerova, C. Lavarino, A. M. Carcaboso, J. Mora, R. Mylvaganam, C. C. Luo, A. Peyrl, M. Popović, A. Azizi, T. T. Batchelor, M. P. Frosch, M. Martinez-Lage, M. W. Kieran, P. Bandopadhyay, R. Beroukhim, G. Fritsch, G. Getz, O. Rozenblatt-Rosen, K. W. Wucherpfennig, D. N. Louis, M. Monje, I. Slavic, K. L. Ligon, T. R. Golub, A. Regev, B. E. Bernstein, M. L. Suvà, Developmental and oncogenic programs in H3K27M gliomas dissected by single-cell RNA-seq, *Science* **360**, 331–335 (2018).
9. E. Rangel-Huerta, E. Maldonado, Transit-Amplifying Cells in the Fast Lane from Stem Cells towards

Differentiation, *Stem Cells Int* **2017**, 7602951–10 (2017).

10. B. Zhang, Y.-C. Hsu, Emerging roles of transit-amplifying cells in tissue regeneration and cancer, *Wiley Interdiscip Rev Dev Biol* **6**, e282 (2017).

11. A. Puissant, S. M. Frumm, G. Alexe, C. F. Bassil, J. Qi, Y. H. Chanthery, E. A. Nekritz, R. Zeid, W. C. Gustafson, P. Greninger, M. J. Garnett, U. McDermott, C. H. Benes, A. L. Kung, W. A. Weiss, J. E. Bradner, K. Stegmaier, Targeting MYCN in Neuroblastoma by BET Bromodomain Inhibition, (2013).

12. J. E. Delmore, G. C. Issa, M. E. Lemieux, P. B. Rahl, J. Shi, H. M. Jacobs, E. Kastritis, T. Gilpatrick, R. M. Paranal, J. Qi, M. Chesi, A. C. Schinzel, M. R. McKeown, T. P. Heffernan, C. R. Vakoc, P. L. Bergsagel, I. M. Ghobrial, P. G. Richardson, R. A. Young, W. C. Hahn, K. C. Anderson, A. L. Kung, J. E. Bradner, C. S. Mitsiades, BET bromodomain inhibition as a therapeutic strategy to target c-Myc, *Cell* **146**, 904–917 (2011).

13. W. Yeo, J. Gautier, Early neural cell death: dying to become neurons, *Dev. Biol.* **274**, 233–244 (2004).

14. D. Szklarczyk, A. Franceschini, S. Wyder, K. Forslund, D. Heller, J. Huerta-Cepas, M. Simonovic, A. Roth, A. Santos, K. P. Tsafou, M. Kuhn, P. Bork, L. J. Jensen, C. von Mering, STRING v10: protein-protein interaction networks, integrated over the tree of life, *Nucleic Acids Res* **43**, D447–52 (2015).

15. X. M. He, C. J. Wikstrand, H. S. Friedman, S. H. Bigner, S. Pleasure, J. Q. Trojanowski, D. D. Bigner, Differentiation characteristics of newly established medulloblastoma cell lines (D384 Med, D425 Med, and D458 Med) and their transplantable xenografts, *Lab. Invest.* **64**, 833–843 (1991).

16. S. Mukherjee, R. Brulet, L. Zhang, J. Hsieh, REST regulation of gene networks in adult neural stem cells, *Nat Commun* **7**, 13360 (2016).

17. M. L. Cook Sangar, L. A. Genovesi, M. W. Nakamoto, M. J. Davis, S. E. Knobluagh, P. Ji, A. Millar, B. J. Wainwright, J. M. Olson, Inhibition of CDK4/6 by Palbociclib Significantly Extends Survival in Medulloblastoma Patient-Derived Xenograft Mouse Models, *Clin. Cancer Res.* **23**, 5802–5813 (2017).

18. S. L. Whiteway, P. S. Harris, S. Venkataraman, I. Alimova, D. K. Birks, A. M. Donson, N. K. Foreman, R. Vibhakar, Inhibition of cyclin-dependent kinase 6 suppresses cell proliferation and enhances radiation sensitivity in medulloblastoma cells, *J. Neurooncol.* **111**, 113–121 (2013).

19. A. R. Hanaford, T. C. Archer, A. Price, U. D. Kahlert, J. Maciaczyk, G. Nikkhah, J. W. Kim, T. Ehrenberger, P. A. Clemons, V. Dančák, B. Seashore-Ludlow, V. Viswanathan, M. L. Stewart, M. G. Rees, A. Shamji, S. Schreiber, E. Fraenkel, S. L. Pomeroy, J. P. Mesirov, P. Tamayo, C. G. Eberhart, E. H. Raabe, DiSCoVERing Innovative Therapies for Rare Tumors: Combining Genetically Accurate Disease Models with In Silico Analysis to Identify Novel Therapeutic Targets, *Clin. Cancer Res.* **22**, 3903–3914 (2016).

20. S. Liao, O. Maertens, K. Cichowski, S. J. Elledge, Genetic modifiers of the BRD4-NUT dependency of NUT midline carcinoma uncovers a synergism between BETis and CDK4/6is, *Genes Dev* (2018), doi:10.1101/gad.315648.118.

21. S. Bolin, A. Borgenvik, C. U. Persson, A. Sundström, J. Qi, J. E. Bradner, W. A. Weiss, Y.-J. Cho, H. Weishaupt, F. J. Swartling, Combined BET bromodomain and CDK2 inhibition in MYC-driven medulloblastoma, *Oncogene* **37**, 2850–2862 (2018).

22. C. I. BLISS, THE TOXICITY OF POISONS APPLIED JOINTLY1, *Annals of Applied Biology* **26**, 585–615 (2008).

Review Figure 1: TUJ1 westerns in mNSCs expressing ORF indicated.

REVIEWERS' COMMENTS:

Reviewer #1 (Remarks to the Author); expert in genetics and medulloblastoma:

I think the authors have done an excellent job of responding to the review. I believe the manuscript is ready for publication and that it will substantially advance our understanding of BET inhibition and resistance in medulloblastoma and possibly, other cancers. I have two minor suggested edits.

Abstract: "...cell cycle regulators as genes mediators of BETi's response." needs an edit.

Results 1st paragraph: "In this way, we systematically evaluated genes suppressed by BETi for both sufficiency (through the CRISPR screen) and necessity (ORF screen) in mediating the BETi response." This sentence seems reversed with regard to sufficiency and necessity or at least seems to be in conflict with the sentence above. I think I understand what is meant but some clarification of the language might be helpful.

Reviewer #2 (Remarks to the Author); expert in genetics and medulloblastoma:

The authors have satisfactorily addressed my comments.

Reviewer #3 (Remarks to the Author); expert in Medulloblastoma Neuronal Growth and Differentiation:

The authors have done an excellent job addressing the issues I and the other reviewers raised in the previous round of review. I have no additional comments or concerns. This is an important and timely study, and I look forward to seeing it published.

We thank all three Reviewers for their time in reviewing our manuscript and for their critiques which we feel have helped improve our manuscript.

Reviewer 1 had two minor edits which we have addressed as below:

1. *'Abstract: "...cell cycle regulators as genes mediators of BETi's response." needs an edit.'*

We have accepted the Editors suggested changes to the abstract, as below:

'BET-bromodomain inhibition (BETi) has shown pre-clinical promise for MYC-amplified medulloblastoma. However, the mechanisms for its action, and ultimately for resistance, have not been fully defined. Here, using a combination of expression profiling, genome-scale CRISPR/Cas9-mediated loss of function and ORF/cDNA driven rescue screens, and cell-based models of spontaneous resistance, we identify bHLH/homeobox transcription factors and cell-cycle regulators as key genes mediating BETi's response and resistance. Cells that acquire drug tolerance exhibit a more neuronally differentiated cell-state and expression of lineage-specific bHLH/homeobox transcription factors. However, they do not terminally differentiate, maintain expression of CCND2, and continue to cycle through S-phase. Moreover, CDK4/CDK6 inhibition delays acquisition of resistance. Therefore, our data provide insights about the mechanisms underlying BETi effects and the appearance of resistance and support the therapeutic use of combined cell-cycle inhibitors with BETi in MYC-amplified medulloblastoma.'

2. *'Results 1st paragraph: "In this way, we systematically evaluated genes suppressed by BETi for both sufficiency (through the CRISPR screen) and necessity (ORF screen) in mediating the BETi response." This sentence seems reversed with regard to sufficiency and necessity or at least seems to be in conflict with the sentence above. I think I understand what is meant but some clarification of the language might be helpful.'*

We have edited this sentence:

'In this way, we systematically evaluated which BETi target genes are both required for cellular proliferation, and able to rescue BETi phenotypes. We considered genes that were nominated by all three assays to be responsible for BETi-induced reductions in cell viability.'